# A Stable, Efficient, and High-Precision Non-Coplanar Calibration Method: Applied for Multi-Camera-Based Stereo Vision Measurements

**DOI:** 10.3390/s23208466

**Published:** 2023-10-14

**Authors:** Hao Zheng, Fajie Duan, Tianyu Li, Jiaxin Li, Guangyue Niu, Zhonghai Cheng, Xin Li

**Affiliations:** 1State Key Laboratory of Precision Measurement Technology and Instruments, Tianjin University, Tianjin 300072, China; zhenghao075_tju@tju.edu.cn (H.Z.); litianyu@tju.edu.cn (T.L.); jiaxinlee@tju.edu.cn (J.L.); niuguangyue@tju.edu.cn (G.N.); chzhh@tju.edu.cn (Z.C.); 2China North Engine Research Institute, Tianjin 30040, China; bjtu_lx@163.com

**Keywords:** camera calibration, non-coplanar calibration method, circular centers feature extraction, monocular and binocular sensing systems, stereo-DIC measurements

## Abstract

Traditional non-coplanar calibration methods, represented by Tsai’s method, are difficult to apply in multi-camera-based stereo vision measurements because of insufficient calibration accuracy, inconvenient operation, etc. Based on projective theory and matrix transformation theory, a novel mathematical model is established to characterize the transformation from targets’ 3D affine coordinates to cameras’ image coordinates. Then, novel non-coplanar calibration methods for both monocular and binocular camera systems are proposed in this paper. To further improve the stability and accuracy of calibration methods, a novel circular feature points extraction method based on region Otsu algorithm and radial section scanning method is proposed to precisely extract the circular feature points. Experiments verify that our novel calibration methods are easy to operate, and have better accuracy than several classical methods, including Tsai’s and Zhang’s methods. Intrinsic and extrinsic parameters of multi-camera-systems can be calibrated simultaneously by our methods. Our novel circular feature points extraction algorithm is stable, and with high precision can effectively improve calibration accuracy for coplanar and non-coplanar methods. Real stereo measurement experiments demonstrate that the proposed calibration method and feature extraction method have high accuracy and stability, and can further serve for complicated shape and deformation measurements, for instance, stereo-DIC measurements, etc.

## 1. Introduction

Camera calibration is the necessary process to determine the unknown basic parameters of camera imaging and the transformation parameters from world coordinate system to camera coordinate system. The mapping relationship between the 3D world coordinates of targets’ feature points and the 2D image coordinates can be used to acquire unknown parameters based on an ideal camera imaging model [1]. Precise calibration parameters of a camera-based measurement system are the prerequisite for the image-based 3D reconstruction, because the calibration parameters directly participate in the mapping progress from 3D reference coordinates to 2D image coordinates or the remapping progress [2,3]. Therefore, developing a high-precision camera calibration method is of great significance.

According to the geometry characteristics of the calibration targets, the existing camera calibration methods can be divided into the following three categories: 3D stereo target calibration methods, 2D planar target calibration methods, and self-calibration methods.

The representative traditional 3D stereo methods are the Direct Linear Transformation (DLT) method developed by Abdel-Aziz [4] and the “Two-Stage” Non-coplanar method developed by Tsai [5]. DLT bridges the gap between photogrammetry and computer vision. Shi [6] developed DLT methods and proposed a DLT-Lines method wherein the camera’s intrinsic and extrinsic parameters can be extracted from the matrix linearly and operate non-linear optimization with distortion coefficients. Tsai gave a two-step calibration method based on radial alignment constraint (RAC). Tsai’s method has a moderate amount of calculation and high accuracy. But the implementation of Tsai’s non-coplanar method is cumbersome. The extrinsic parameters calibrated by Tsai’s method are inaccurate and the tangential distortion parameters of lenses cannot be calibrated by Tsai’s method because of the RAC model’s insufficiency. J. Zhang [7] and Zheng [8] et al. noticed the drawbacks of Tsai’s method and they gave some corrective action for Tsai’s non-coplanar methods. However, their research is still not impeccable and cannot efficiently apply in multi-camera calibration tasks. The deficiencies of Tsai’s method and some incomplete improvements will be further discussed in Section 2.3.

It is the case that 2D planar target calibration methods are also called coplanar calibration methods [5,8,9,10,11]. The representative coplanar method was proposed by Zhang [9]. As a milestone in camera calibration, Zhang’s method is easy to use in practical calibration and provides sufficient accuracy for most applications. Zhang’s method needs to take images of 2D targets in multi-viewing multiplane position, and then calculate the intrinsic and extrinsic parameters and distortion coefficients through linearly initial parameters solving and non-linear optimization. Zhu et al. [10], Sels et al. [11], Chen et al. [12], and many other scholars further developed Zhang’s method. Most of the listed scholars made improvements by changing the calibration patterns, increasing the numbers of targets’ feature points, and increasing the extraction accuracy of feature points in images. It must be admitted that the calibration accuracy and uncertainty of calibrated parameters are accorded certain promotions by the above scholars’ research. However, the above promotions are mostly established at the expense of applying complicated time-consuming feature extraction algorithms. Meanwhile, there is rarely limited innovation in the calibration model and mathematical operation process of Zhang’s original calibration methods.

Self-calibration methods only use the corresponding relationship between the surrounding images and the images during the movement to perform the calibration. Hartley [13] and Maybank and Faugeras et al. [14] first proposed the idea of camera self-calibration. Due to the unstable characteristics of the nature feature and feature extraction algorithms, self-calibration methods are hard to maintain with high accuracy and robustness. Li et al. [15] and Li et al. [16] and other scholars tried to use manually selecting features to replace nature textural features. Calibration accuracy is relatively improved at the huge expanse of time-consuming feature extraction and increasing mathematical complexity. In general, existing self-calibration methods have poor accuracy, low efficiency, and low robustness, so that they are difficult to use in high-precision stereo measurements. This paper focus more on developing a stable, efficient, and accurate calibration method for multi-camera-based high-precision stereo measurements. As a result, we will not discuss the self-calibration method in the rest of this paper.

Camera-sensing-systems which are used for stereo measurements can be roughly divided into two types. The first type is a monocular camera measurement system. A single camera cannot directly acquire spatial depth information, but a monocular camera system can often contain extra feature projection devices, e.g., line-structured light camera-based sensors, laser light structure camera-based sensors, etc. Calibration for structured-light-based monocular camera measurement systems mainly include two progresses, which are intrinsic parameters calibration for the monocular camera and the extrinsic parameters calibration for the connecting structure of the monocular camera and light projection device. Calibration for structured-light-based monocular camera measurement has been studied by many scholars [17,18]. This paper focus more on the calibration of the second camera sensing system, which is the multi-camera-based stereo measurement system. A binocular camera system is the most typical and fundamental multi-camera system. The calibration of a binocular camera system also contains two steps, which are separate intrinsic parameters calibration for two single cameras, and the extrinsic structural parameters’ calibration of these two cameras.

Camera-based stereo measurements are built on the 3D reconstruction of features from images. The 3D reconstruction of image features mainly contains three steps, i.e., features’ extraction, features’ stereo matching, and features’ stereo reconstruction based on triangulation. There are different suitable feature extraction and matching methods for different kinds of image features. For the discrete and limited artificial geometrical features, i.e., points, circles, corners, lines, etc., scholars use specific geometrical features extraction algorithms to extract and match the features. For the local features with stubborn scale, orientation, and illumination invariance, scholars use local feature detectors and descriptors, e.g., scale invariant feature transform (SIFT) [19], speeded up robust features (SURF) [20], oriented FAST and rotated BRIEF (ORB) [21], etc., which have been introduced to distract, descript, and match the related local key points. For the full-field measurements of surface shape, displacement, and deformation, etc., if we have lower demands of measurements accuracy, then Block Matching (BM), Semi-Global Block Matching (SGBM), Graph Cut (GC), and Dynamic Programming (DP), etc., can be used to perform global matching. If we have higher demands of measurements accuracy, then digital image correlation (DIC) has been demonstrated as one of the most effective technologies to quantitatively extract full-field displacement and strain responses of materials, components, structures, or biological tissues through correlation-based matching strategies [22].

In this paper, we propose a novel stable, efficient, and accurate non-coplanar calibration method which can be well applied for camera-based stereo vision measurements. This method can significantly correct existing non-coplanar calibration methods’ weaknesses, i.e., cumbersome operation, insufficient accuracy, unstable fitting of different measurement scenes, etc. Specifically, the contributions of this study are summarized as follows: This paper establishes a novel improved affine coordinates correction mathematical model for non-coplanar calibration. A novel calibration method based on this model is established for both monocular and binocular camera systems. Simulation and real experiments verified that our novel methods have better accuracy, stability, and efficiency than controlled methods.For further improving the accuracy and stability of existing calibration methods, a novel simple circle feature points extraction algorithm based on the combining of local OTSU and gradient-based radial section scanning for edges is proposed in this paper. Simulations and real experiments demonstrated our algorithm has better performance in extraction accuracy and stability for illumination and viewing angle changes than the traditional algorithm from OpenCV.Real all-process 3D reconstruction experiments of both discrete feature points and object surface’s full-field region of interest (ROI) have been operated from the stereo system’s calibration, features extraction, features’ stereo matching, to the final features’ stereo reconstruction. Experiments demonstrate the feasibility of our calibration methods for real measurement scenes, and the stereo measurements with this paper’s calibration parameters have better accuracy than controlled methods.

The layout of this paper is organized as follows. Section 2 formulates related works, models, strategies, and restrictions of current camera calibration methods and stereo measurements. Section 3 and Section 4 present the methodology of this paper’s research. The experiments and results are presented in Section 5. Finally, Section 6 concludes the paper and indicates future directions.

## 2. Related Works and Problems Formulation

### 2.1. Mathematical Model and Some Developments of Tsai’s Non-Coplanar Calibration Method

We have summed up Tsai’s non-coplanar mathematical model [5] in Table 1. Based on radial alignment constraint (RAC), an overdetermined equation can be used to solve seven independent intermediate variables $\left[\begin{array}{cccc} a_{1} & a_{2} & \cdots & a_{7} \end{array}\right]$. Then, with two orthogonal constraints of rotation constrains, $\left[\begin{array}{cccc} s_{x} & R_{3\times 3} & T_{x} & T_{y} \end{array}\right]$ can be solved first. Correspondingly, the initial values of $\left[\begin{array}{cc} T_{z} & f \end{array}\right]$ can be solved linearly with calculated parameters and another overdetermined equation. At last, use nonlinear optimization to refine part of the calibration parameters and radial distortion coefficients. The calibration with Tsai’s method for a binocular camera system is the combination of two separate calibrations for single cameras.

Tsai’s calibration method has a simple mathematical model and it is very easy to operate in an algorithm. The algorithm has good efficiency for most of the parameters are computed by linear overdetermined equations and very few elements are brought into non-linear optimization. However, Tsai’s calibration mathematical model has some drawbacks. Firstly, RAC constraints take no account of the tangential distortion of the lens. J. Zhang et al. [7] and Xu et al. [23] introduced the tangential distortion model into Tsai’s method and significantly improved the calibration accuracy. Tang et al. [24] also considered tangential distortion with Tsai’s model and verified that Tsai’s method has better efficiency in an algorithm than Zhang’s method. 

Despite the better efficiency of Tsai’s method, there are fewer scholars and engineers applying Tsai’s method in real measurements than there are applying Zhang’s method. This is because Tsai’s method is less about accuracy and is inconvenient to implement which can be seen in detail in Section 2.3 and Section 2.5.

### 2.2. Mathematical Model and Recent Development of Coplanar Calibration Method

As a milestone of calibration methods, Zhang’s coplanar calibration method has been applied in many computer vision tasks for its convenience, accuracy, and stability. The mathematical model in [9] is summarized in Table 2.

Using ${\tilde{P}}_{I}$ and ${\tilde{P}}_{W}$ respectively represents the matched homogeneous coordinates from the 2D image pixel coordinate system and world coordinate system. The calibration process of coplanar methods based on the transformation from 2D world coordinates to 2D image pixel coordinates, which can be described as an 3 × 3 homography matrix, is shown as ***H***_3×3_ in Table 2. However, one ***H***_3×3_ matrix can only supply two constraints for the linear solution of parameters, yet there are five intrinsic parameters to be solved. Thus, at least two images from different orientations are needed to evaluate the four initial values of intrinsic parameters if impose *γ* = 0. Then, the initial intrinsic parameters are used to calculate the rotation and translation vectors between each planar target’s world coordinate system and camera system. It is worth mentioning that if the initial ***K_Zhang_*** and ***H***_3×3_ are directly used to compute ***R***_3×3_, the ***R***_3×3_ cannot strictly satisfy the orthogonal properties of a rotation matrix. Singular value decomposition is used to approximate the relatively best rotation matrix in [9]. Then, ***R***_3×3_ rotation matrices are transferred to ***R***_3×1_ rotation vectors. For the DOF of a certain rotation is three, obviously, ***R***_3×1_ rotation vectors can better meet the demands of follow-up nonlinear optimization. ***R***_3×1_ and ***R***_3×3_ are related by the Rodrigues formula. Using ***R***_3×1_ vector rather than ***R***_3×3_ matrix in nonlinear optimization can avoid the problem of insufficient orthogonality of a certain rotation matrix.

Recently, scholars have developed different calibration methods based on Zhang’s calibration model. Yin et al. [25] improved the binocular calibration accuracy by timing correction of two consecutive frames; Cheng et al. [26] used the perspective correction and phase estimation method together to help increase the accuracy of control point localization and consequently of camera calibration; Chen et al. [27] applied sub-pixel edge detection and cross ratio invariance to refine the circular control points’ image position and then increase the accuracy of calibration; Wang et al. [28] extended Pascal’s theorem to the affine plane to obtain the constraints of the circular points in images and used properties of circle and infinite line to calibrate the intrinsic parameters of camera; Dong et al. [29] developed a confidence-based camera calibration method with modified census transform for chessboard patterns whereby this method is effective to achieve accurate calibration results; Zhang et al. [30] designed particular stereo targets with multiple feature planes to help simultaneously identify the intrinsic and extrinsic parameters of a camera system by a single captured image.

Obviously, scholars focus more on the improvements of feature points extraction methods and accuracy rather than the mathematical model of coplanar methods. This research tendency partly reflects the accuracy of Zhang’s calibration model which has been confirmed effective by most scholars. The improvements nonetheless can only emerge in other aspects except the mathematical model.

### 2.3. Deficiencies of Tsai’s Calibration Model and Recent Research of Non-Coplanar Calibration Model Based on Affine Coordinate Correction

Zheng et al. [8] pointed out how the uncorrected sliding direction of planar calibration target would greatly influent the accuracy of Tsai’s non-coplanar method. This inference can be confirmed by the simulation in Figure 1. Assume there is an uncorrected yaw angle between sliding direction and ideal world coordinate system decided by planar target and its normal vector direction. If the sliding shifts are erroneously assumed to be the *z_w_* in a world system, using Tsai’s method will obtain the results in Figure 1. The absolute value of focal length’s relative error increases to 20.47% when the yaw angle between sliding direction and planar targets increases to 4°, and they will continue increasing along with the increasing angle.

Moreover, Zheng et al. [8] also pointed out that Tsai’s mathematical model cannot obtain a strictly orthogonal rotation matrix, for the following reason: The 1st step of linear initial values solving uses seven independent intermediate variables to acquire $\left[\begin{array}{cccc} s_{x} & R_{3\times 3} & T_{x} & T_{y} \end{array}\right]$ of six DOF. This is essentially an overdetermined equation solving procedure and can only obtain an approximate solution. Assume ***r***1, ***r***2 are the first two rows of ***R***_3×3_. In this procedure, Tsai can only use the two single inner product properties of ***r***1 and ***r***2 to calculate *s_x_*, *T_y_*. If one needs to obtain an orthonormal matrix by this way, the other property of ***r***1 × ***r***2 = 0 should be applied to ensure the orthogonality of ***R***_3×3_. However, Tsai’s method missed this constraint and did not make any compensation in the following nonlinear optimization. 

Zheng et al. [8] proposed a novel non-coplanar method based on an affine coordinate correction (ACC) model as shown in Table 3. This method is called the ACC method in this paper based on mathematical characteristics.

The ACC method introduced a 2D normalized vector $\eta ={\left(\begin{array}{cc} \eta_{x} & \eta_{x} \end{array}\right)}^{T}$ to correct the planar target’s two axis, and a 3D normalized vector $\beta ={\left(\begin{array}{ccc} \beta_{x} & \beta_{y} & \beta_{z} \end{array}\right)}^{T}$ to correct the sliding direction vertical to the planar. The optical center is assumed fixed at the center of the image. Since the number of intermediate parameters is equal to the DOF of parameters to be calibrated, take 11 intermediate parameters solved by overdetermined equations into nonlinear optimization together with distortion coefficients. With enough orthonormal constraints of the rotation matrix and the properties of normalized vectors, the final parameters can easily obtain analytical solutions from optimized intermediate parameters.

The ACC calibration model works well and can obtain accurate calibration results for a monocular camera system. However, this model cannot fit the calibration well for a binocular camera system as shown in Table 4. 

Firstly, Table 3 illustrates that the DOF of the intermediate parameters of a single camera is 11. If the ACC model is extended to a binocular camera system such as in Table 4, one will obtain two separate intermediate matrices of which the sum DOF is 22. But the physical meanings of ***η*** and ***β*** illustrate that they should be equal for each single camera when operating a non-coplanar calibration for a binocular system. This means that the DOF of the parameters to be calibrated reduces to 19 and the original unconstrained nonlinear optimization of the intermediate parameters cannot apply for the binocular system. Constraints from the rotation matrix and correction vectors were introduced to be the penalty constraints and construct nonlinear optimization. To further simplify the binocular calibration model based on the ACC model and maintain the orthogonality of the rotation matrix, Zheng et al. [8] chose to calibrate the intrinsic parameters for the single camera first in order to reduce the DOF of the parameters to be calibrated and introduce enough penalty constraints. 

Unfortunately, this mathematical compromise did not bring an advance of calibration accuracy but rather introduced an extra workload for determining the penalty coefficients of each penalty constraint.

To overcome the above problems either existing in Tsai’s method or in the ACC method, this paper proposes a novel improved affine coordinate correction (IACC) calibration method for both monocular and binocular camera systems.

### 2.4. Local Spatial Optimality of Calibration Parameters

No matter what method is used to evaluate the accuracy of calibration parameters, the optimality of these parameters is only mathematically optimal, and approximately physically l optimal. The approximation properties make calibration parameters lack strictly physical significance. This means that the calibration parameters from different calibration methods must have optimum properties in particular time and space domains.

The above analysis indicates that measurement objects which occur at different positions within the FOV of camera systems have their own optimal calibration parameters for the best measurement results. Experience suggests that the more calibration targets cover the measurement position, the more fitting calibration parameters can be acquired for measurements. These deductions can be reflected in Figure 2. This kind of local optimality may also happen in domains of time. Compared to the spatial optimality, current camera sensors’ hardware exhibits more stability in a short time interval. Thus, this paper focuses more on the local spatial optimality of calibration parameters. Experiments in Section 5.5 can support these deductions about the local spatial optimality of calibration parameters.

The depth of field (DOV) within FOV of camera system can be roughly evaluated by the nominal value of lens’ focal length, aperture, allowable dispersion circle’s diameter, and object distance. Then, measurement objects can be placed at the range of DOV. If we want to acquire the best measurement accuracy, calibration targets’ feature points should come close to the measurement position and cover the limited measurement depth as much as possible. It is worth noting that the greater covered depth of planar targets is not better than the accurate smaller covered depth decided by measured objects’ actual 3D information, especially for high precision close-range photogrammetry. 

### 2.5. Implementation and Restrictions of Coplanar Calibration Methods and Non-Coplanar Calibration Methods

As shown in Figure 3a, coplanar methods for monocular camera systems need to take images of planar targets from various orientations (at least two). If planar targets were set in mono-viewing positions, coplanar methods would lose efficacy. As shown in Figure 3b, non-coplanar methods need to take images of planar targets from one fixed orientation. At least two images and one known shift of targets are needed to carry out a continued calibration process. Correspondingly, the implementation process of binocular camera system calibration by coplanar methods and non-coplanar methods are shown in Figure 3c,d. Basically, non-coplanar method is an improved calibration method using virtual 3D stereo targets. For guaranteeing the geometric accuracy of patterns’ world coordinates, extra equipment (Tsai’s method) or mathematical model (ACC and IACC methods) should be introduced to make the correction of world coordinates. This may bring in extra workloads, but the mathematical method is obviously more efficient than the manual adjustment method with extra instruments.

In most actual measurements, the rough 3D information of measured objects is known, and the structure of a multi-camera measurement system is specially designed for the measured object. If the depth’s changing range is not too large, it is better to perform calibration over the depth’s changing range than over the whole FOV. On this occasion, non-coplanar calibration methods are more applicable than coplanar methods, for tiny shifts of planar targets in one fixed direction could generate large amounts of feature points for calibration tasks. On the contrast, considering the size of planar targets and the demands of multi-viewing images, the equivalent amount of feature points needs larger depth range than measured objects’ depth range on most occasions. Otherwise, minor inclination angles’ change of planar targets may not support coplanar methods to obtain the right calibration parameters. Thus, the restriction of coplanar methods comes from the demands of image acquisition from different viewing angles, and the restriction of non-coplanar methods comes from the measurement and correction of sliding shifts. 

### 2.6. Strategies of Enhancing Calibration Performance by Improving Feature Points Extraction Algorithms

Alongside calibration models, the improvements of calibration targets’ 3D production and corresponding 2D image feature extraction can directly enhance calibration results. 

The introduction of scholars’ strategies to improve the calibration performance by enhancing 2D image feature extraction algorithm is put in Appendix A. 

This paper concentrates more on the improvements of extracting accuracy and stability for symmetric circles. A novel extraction algorithm for symmetric circles’ pattern is proposed in this paper, which has better extraction accuracy, stability of illumination, and targets’ orientation changes than OpenCV’s traditional algorithm. Correspondingly, the performance of calibration accuracy and stability can further be enhanced by using the proposed algorithm.

## 3. Novel Calibration Mathematical Model

### 3.1. Present Novel Improved Affine Coordinate Correction Mathematical Model for Non-Coplanar Calibration

With the analysis in Section 2.1, Section 2.2, Section 2.3 and Section 2.4, a novel improved affine coordinate correction (IACC) mathematical model for non-coplanar calibration is proposed. As shown in Table 5, using ${\tilde{P}}_{I}$ and ${\tilde{P}}_{p}$ respectively represent the matched homogeneous coordinates of the 2D image pixel coordinate system and 3D affine coordinate system built from uncorrected planar target’s two axes and 1D sliding direction.

### 3.2. Coordinate Space Transformation from Target Affine Space to Orthogonal World Coordinate Space

Figure 4’s left part shows the corrections for the calibration target’s affine coordinates from the ACC model, in which normalized 2D vector ***η*** corrects the planar target’s vertical axis and horizontal axis, and in which normalized 3D vector ***β*** corrects the sliding direction into orthogonal world coordinate system *O_w_-X_w_Y_w_Z_w_*. In this case, ***η*** and ***β*** are introduced to describe the skews of the planar target’s two axes and stage’s sliding direction, if the planar target and sliding direction remain fixed, $\eta ={\left(\eta_{x},\eta_{y}\right)}^{T}$ and $\beta ={\left(\beta_{x},\beta_{y},\beta_{z}\right)}^{T}$ should keep unchanged. However, when we set calibration experiments with the ACC method for different cameras with the same sliding stage and planar target (remain fixed), ***η*** and ***β*** do not always stay the same. The change of ***η*** is more remarkable than ***β*** with the switch of calibrating different cameras. It is more likely that the 1 DOF from ***η*** should transfer to characterize some intrinsic properties of different cameras. In fact, our planar targets are fabricated with optical glass and high precision (close to 1 μm) lithography process, which means ***η*** should be infinitely close to ${\left(0,1\right)}^{T}$.

Based on the actual physical reality, we propose our novel improved affine coordinate correction model as shown in Figure 4’s right part. In our calibration model, the planar target’s two axes are considered strictly orthogonal. This means original ***η*** should be adjusted to ${\left(0,1\right)}^{T}$. As shown in Figure 4’s right part, *O_w_-X_w_Y_w_Z_w_* is the ideal orthogonal 3D world coordinate system of the planar target. Normalized 3D vector ***β*** remains in our calibration model to correct the sliding direction to the ideal.

Correspondingly, the transformation equation should be adjusted as follow: (1)$$\left[\begin{array}{c} X_{w}\\ Y_{w}\\ Z_{w} \end{array}\right]=\left[\begin{array}{ccc} 1 & \eta_{x} & \beta_{x}\\ 0 & \eta_{y} & \beta_{y}\\ 0 & 0 & \beta_{z} \end{array}\right]\left[\begin{array}{c} X_{p}\\ Y_{p}\\ Z_{p} \end{array}\right]\to \left[\begin{array}{c} X_{w}\\ Y_{w}\\ Z_{w} \end{array}\right]=\left[\begin{array}{ccc} 1 & 0 & \beta_{x}\\ 0 & 1 & \beta_{y}\\ 0 & 0 & \beta_{z} \end{array}\right]\left[\begin{array}{c} X_{p}\\ Y_{p}\\ Z_{p} \end{array}\right]$$

Figure 5 shows the pinhole imaging model of the camera. *O-XYZ* is the camera coordinate system, of which the unit is mm. *O_R_-UV* is the camera image sensor’s two-dimensional pixel coordinate system, of which the origin point is in the upper left corner of the image sensor and the unit is Pixel. *O_u_-x_u_y_u_* is the image-plane coordinate system, of which the unit is mm.

According to the theory of rigid transformation, the transformation relationship between the camera coordinate system *O-XYZ* and the calibration target object world coordinate system *O_w_-X_w_Y_w_Z_w_* can be expressed as:(2)$$\left[\begin{array}{c} X\\ Y\\ Z \end{array}\right]=R\left[\begin{array}{c} X_{w}\\ Y_{w}\\ Z_{w} \end{array}\right]+T$$

In which $R=\left[\begin{array}{ccc} r_{1} & r_{2} & r_{3}\\ r_{4} & r_{5} & r_{6}\\ r_{7} & r_{8} & r_{9} \end{array}\right]$, and $T=\left[\begin{array}{c} T_{x}\\ T_{y}\\ T_{z} \end{array}\right]$.

The transformation of points’ coordinates between system *O_R_-UV* and *O-XYZ* can be expressed by Equation (3):(3)$$\rho {\tilde{P}}_{I}=\rho \left[\begin{array}{l} U\\ V\\ 1 \end{array}\right]=\left[\begin{array}{ccc} f_{x} & \gamma & U_{0}\\ 0 & f_{y} & V_{0}\\ 0 & 0 & 1 \end{array}\right]\left[\begin{array}{l} X\\ Y\\ Z \end{array}\right]$$

Focal length in (3) is expressed as *f_x_* and *f_y_*, which separately express the focal length’s equivalent pixel numbers in the sensor’s horizontal and vertical direction. For the use of homogeneous coordinates transformation, *ρ* in this paper denotes the proportionality coefficients of transformation and have no strict physical meanings. 

Thus, the ideal process from the affine space coordinate system *O_p_-X_p_Y_p_Z_p_* to the camera image sensor’s two-dimensional pixel coordinate system *O_R_-UV* can be expressed as:(4)$$\rho ({\tilde{P}}_{I}-{\tilde{P}}_{c})=\rho \left[\begin{array}{l} U-U_{0}\\ V-V_{0}\\ 1 \end{array}\right]=\left[\begin{array}{ccc} f_{x} & \gamma & 0\\ 0 & f_{y} & 0\\ 0 & 0 & 1 \end{array}\right]\left[\begin{array}{cccc} r_{1} & r_{2} & r_{3} & T_{x}\\ r_{4} & r_{5} & r_{6} & T_{y}\\ r_{7} & r_{8} & r_{9} & T_{z} \end{array}\right]\left[\begin{array}{cccc} 1 & 0 & \beta_{x} & 0\\ 0 & 1 & \beta_{y} & 0\\ 0 & 0 & \beta_{z} & 0\\ 0 & 0 & 0 & 1 \end{array}\right]\left[\begin{array}{l} X_{p}\\ Y_{p}\\ Z_{p}\\ 1 \end{array}\right]=A_{3\times 4}{\tilde{P}}_{p}$$

Compared with the ACC model in Table 3, our novel model IACC introduces *γ* to describe the skewness of the image sensor’s two axes. If the actual included angle of the imaging sensor’s two axes is *θ*, there is $\gamma =f_{y}\cdot \cot\theta$ in physical meaning and the *f_y_* in (4) illustrates $f_{y}=f_{y}/\sin\theta$. The physical meaning of *γ* illustrates that when *θ* is close to 90°, *γ* should be close to 0. Thus, many scholars and engineers choose to regard *γ* as 0 in actual scenes based on the manufacturing level of current industrial cameras.

But what really appeals to us is that *γ* can supply 1 DOF for the intermediate matrix ***A***_3×4_, which we lost when we abandon ***η***, and *γ* is an intrinsic parameter for a camera. We will further give our explanation about the significance of *γ* for IACC in Section 4.1.

### 3.3. Processing of Lens’ Distortion

There are two main types of distortion errors for lens due to the inevitable processing and assembly error, i.e., radial distortion and tangential distortion. 

The radial distortion is symmetrical about the main optical axis of the camera, and its mathematical model can be expressed as:(5)$$\left\{ \begin{array}{l} \delta_{xr}{}^{\prime }=x_{u}{}^{\prime }\left(k_{1}q^{2}+k_{2}q^{4}+k_{3}q^{6}+\cdot \cdot \cdot \right)\\ \delta_{yr}{}^{\prime }=y_{u}{}^{\prime }\left(k_{1}q^{2}+k_{2}q^{4}+k_{3}q^{6}+\cdot \cdot \cdot \right) \end{array}\right.$$

In which, $q=\sqrt{x_{u}{{}^{\prime }}^{2}+y_{u}{{}^{\prime }}^{2}}$, $\left(x_{u}{}^{\prime },y_{u}{}^{\prime }\right)$ expresses the ideal (undistorted) normalized coordinates of the image-plane coordinate system *O_u_-x_u_y_u_* and the distortion center is *O_u_*. *k_1_*, while *k_2_*… are the radial distortion coefficients in which generally only the first two order coefficients play a major role.

The tangential distortion is not symmetrical about the main optical axis of the camera lens, and its mathematical model is:(6)$$\left\{ \begin{array}{l} \delta_{xt}{}^{\prime }=p_{1}\left(q^{2}+2x_{u}{{}^{\prime }}^{2}\right)+2p_{2}x_{u}{}^{\prime }y_{u}{}^{\prime }\\ \delta_{yt}{}^{\prime }=2p_{1}x_{u}{}^{\prime }y_{u}{}^{\prime }+p_{2}\left(q^{2}+2y_{u}{{}^{\prime }}^{2}\right) \end{array}\right.$$

In this formula, *p*_1_ and *p*_2_ express the first two order tangential distortion coefficients.

In the image plane system *O_u_-x_u_y_u_*, the mathematical relationship between the ideal imaging point’s normalized coordinate $\left(x_{u}{}^{\prime },y_{u}{}^{\prime }\right)$ and the actual imaging point’s normalized coordinate $\left(x_{d}{}^{\prime },y_{d}{}^{\prime }\right)$ can be expressed by Equation (9). Note that the subscript *u* in this paper represents the ideal coordinate value, the subscript *d* represents the coordinate value with distortion, the superscript “ ’ ” represents normalized coordinates, and superscript “^”represents ideal image points’ coordinates from reprojection.
(7)$$\left\{ \begin{array}{l} x_{d}{}^{\prime }=x_{u}{}^{\prime }+\delta_{\mathrm{xr}}{}^{\prime }+\delta_{\mathrm{xt}}{}^{\prime }\\ y_{d}{}^{\prime }=y_{u}{}^{\prime }+\delta_{\mathrm{yr}}{}^{\prime }+\delta_{\mathrm{yt}}{}^{\prime } \end{array}\right.$$

Combining Equation (3) with known calibrated parameters, the ideal image points’ coordinates $\left({\hat{U}}_{d},{\hat{V}}_{d}\right)$ in *O_R_-UV* should be expressed as:(8)$$\left\{ \begin{array}{l} {\hat{U}}_{d}=f_{x}x_{d}{}^{\prime }+\gamma y_{d}{}^{\prime }+U_{0}\\ {\hat{V}}_{d}=f_{y}y_{d}{}^{\prime }+V_{0} \end{array}\right.$$

## 4. Key Procedures of IACC Calibration Method 

### 4.1. Initial Value Linear Solving and Parameters Separation Method

The relationship between ***A***_3×4_ from Equation (4) and parameters to be calibrated can be expressed as:(9)$$A_{3\times 4}=\left[\begin{array}{cccc} a_{1} & a_{2} & a_{3} & a_{4}\\ a_{5} & a_{6} & a_{7} & a_{8}\\ a_{9} & a_{1}{}_{0} & a_{11} & 1 \end{array}\right]=\left[\begin{array}{cccc} \frac{f_{x}r_{1}+\gamma r_{4}}{T_{z}} & \frac{f_{x}r_{2}+\gamma r_{5}}{T_{z}} & \frac{f_{x}(\beta_{x}r_{1}+\beta_{y}r_{2}+\beta_{z}r_{3})+\gamma (\beta_{x}r_{4}+\beta_{y}r_{5}+\beta_{z}r_{6})}{T_{z}} & \frac{f_{x}T_{x}+\gamma T_{y}}{T_{z}}\\ \frac{f_{y}}{T_{z}}r_{4} & \frac{f_{y}}{T_{z}}r_{5} & \frac{f_{y}(\beta_{x}r_{4}+\beta_{y}r_{5}+\beta_{z}r_{6})}{T_{z}} & \frac{f_{y}}{T_{z}}T_{y}\\ \frac{1}{T_{z}}r_{7} & \frac{1}{T_{z}}r_{8} & \frac{\beta_{x}r_{7}+\beta_{y}r_{8}+\beta_{z}r_{9}}{T_{z}} & 1 \end{array}\right]$$

In which ***A***_3×4_ can supply 11 DOF. And the DOF of final parameters (the intrinsic and extrinsic parameters, excepted for $P_{c}=(U_{0},V_{0})$) to be calibrated is 11. Theoretically, the analytical solution of the camera’s intrinsic and extrinsic parameters can be solved directly from *a*_1_~*a*_11_ by the algebraic solving method.

Firstly, assume ***P***_c_ is at the image center, solve the initial value of *a*_1_~*a*_11_ by the linear least squares method. With Equations (4) and (9), there are:(10)$$\left\{ \begin{array}{l} U_{i}-U_{0}=\frac{a_{1}X_{pi}+a_{2}Y_{pi}+a_{3}Z_{pi}+a_{4}}{a_{9}X_{pi}+a_{10}Y_{pi}+a_{11}Z_{pi}+1}\\ V_{i}-V_{0}=\frac{a_{5}X_{pi}+a_{6}Y_{pi}+a_{7}Z_{pi}+a_{8}}{a_{9}X_{pi}+a_{10}Y_{pi}+a_{11}Z_{pi}+1} \end{array}\right.$$

With N pairs of corresponding calibration feature points, we can obtain the least squares solution $a={\left[\begin{array}{ccccccccccc} a_{1} & a_{2} & a_{3} & a_{4} & a_{5} & a_{6} & a_{7} & a_{8} & a_{9} & a_{10} & a_{11} \end{array}\right]}^{T}$.

With enough orthogonal constraints and properties of normalized vector ***β*** shown in Equation (11)’s left part, Equation (11)’s right part can be deduced as follows:(11)$$\left\{ \begin{array}{l} r_{1}{}^{2}+r_{4}{}^{2}+r_{7}{}^{2}=1\\ r_{2}{}^{2}+r_{5}{}^{2}+r_{8}{}^{2}=1\\ r_{3}{}^{2}+r_{6}{}^{2}+r_{9}{}^{2}=1\\ r_{1}r_{2}+r_{4}r_{5}+r_{7}r_{8}=0\\ r_{2}r_{3}+r_{5}r_{6}+r_{8}r_{9}=0\\ r_{1}r_{3}+r_{4}r_{6}+r_{7}r_{9}=0\\ \beta_{x}{}^{2}+\beta_{y}{}^{2}+\beta_{z}{}^{2}=1 \end{array}\right.\to \left\{ \begin{array}{l} a_{1}{}^{2}\frac{T_{z}{}^{2}}{f_{x}{}^{2}}+a_{5}{}^{2}(\frac{T_{z}{}^{2}\gamma^{2}+f_{x}{}^{2}T_{z}{}^{2}}{f_{x}{}^{2}f_{y}{}^{2}})-2a_{1}a_{5}(\frac{T_{z}{}^{2}\gamma }{f_{x}{}^{2}f_{y}})+a_{9}{}^{2}T_{z}{}^{2}=1\\ a_{2}{}^{2}\frac{T_{z}{}^{2}}{f_{x}{}^{2}}+a_{6}{}^{2}(\frac{T_{z}{}^{2}\gamma^{2}+f_{x}{}^{2}T_{z}{}^{2}}{f_{x}{}^{2}f_{y}{}^{2}})-2a_{2}a_{6}(\frac{T_{z}{}^{2}\gamma }{f_{x}{}^{2}f_{y}})+a_{10}{}^{2}T_{z}{}^{2}=1\\ a_{3}{}^{2}\frac{T_{z}{}^{2}}{f_{x}{}^{2}}+a_{7}{}^{2}(\frac{T_{z}{}^{2}\gamma^{2}+f_{x}{}^{2}T_{z}{}^{2}}{f_{x}{}^{2}f_{y}{}^{2}})-2a_{3}a_{7}(\frac{T_{z}{}^{2}\gamma }{f_{x}{}^{2}f_{y}})+a_{11}{}^{2}T_{z}{}^{2}=1\\ a_{1}a_{2}\frac{T_{z}{}^{2}}{f_{x}{}^{2}}+a_{5}a_{6}(\frac{T_{z}{}^{2}\gamma^{2}+f_{x}{}^{2}T_{z}{}^{2}}{f_{x}{}^{2}f_{y}{}^{2}})-(a_{1}a_{6}+a_{2}a_{5})(\frac{T_{z}{}^{2}\gamma }{f_{x}{}^{2}f_{y}})+a_{9}a_{10}\cdot T_{z}{}^{2}=0\\ a_{1}a_{3}\frac{T_{z}{}^{2}}{f_{x}{}^{2}}+a_{5}a_{7}(\frac{T_{z}{}^{2}\gamma^{2}+f_{x}{}^{2}T_{z}{}^{2}}{f_{x}{}^{2}f_{y}{}^{2}})-(a_{1}a_{7}+a_{3}a_{5})(\frac{T_{z}{}^{2}\gamma }{f_{x}{}^{2}f_{y}})+a_{9}a_{11}\cdot T_{z}{}^{2}=\beta_{x}\\ a_{2}a_{3}\frac{T_{z}{}^{2}}{f_{x}{}^{2}}+a_{6}a_{7}(\frac{T_{z}{}^{2}\gamma^{2}+f_{x}{}^{2}T_{z}{}^{2}}{f_{x}{}^{2}f_{y}{}^{2}})-(a_{2}a_{7}+a_{3}a_{6})(\frac{T_{z}{}^{2}\gamma }{f_{x}{}^{2}f_{y}})+a_{10}a_{11}\cdot T_{z}{}^{2}=\beta_{y} \end{array}\right.$$

In the calibration process, *T_z_* > 0, *f_x_* > 0, *f_y_* > 0 and $\beta_{z}$ > 0 are specified. The analytical solutions of $\frac{T_{z}{}^{2}}{f_{x}{}^{2}}$, $\frac{T_{z}{}^{2}\gamma^{2}+f_{x}{}^{2}T_{z}{}^{2}}{f_{x}{}^{2}f_{y}{}^{2}}$, $\frac{T_{z}{}^{2}\gamma }{f_{x}{}^{2}f_{y}}$ and $T_{z}{}^{2}$ can be obtained by solving the first four equations in Equation (11). Further, we can solve these four parameters, i.e., *f_x_*, *f_y_*, *γ*, and *T_z_*. $\beta_{x}$ and $\beta_{y}$ can be solved by the Equations (5) and (6). 

In physical meanings, the introduction of *γ* supplies an intrinsic parameter for individual cameras. And in mathematical meanings, *γ* can supply 1 DOF for the intermediate matrix ***A***_3×4_, which we lost when we abandoned ***η***. Since the DOF of the final parameters (the intrinsic and extrinsic parameters, excepted for the DOF of final parameters (the intrinsic and extrinsic parameters, excepted for $P_{c}=(U_{0},V_{0})$)) is equal to the DOF of ***A***_3×4_. The orthogonality of ***R***_3×3_’s analytical solution can be guaranteed. 

Bringing the above parameters back into Equation (9), the remaining extrinsic parameters can be obtained as follows: (12)$$\left\{ \begin{array}{l} r_{4}=\frac{a_{5}T_{z}}{f_{y}}\\ r_{7}=a_{9}T_{z}\\ r_{1}=\frac{a_{1}T_{z}-\gamma r_{4}}{f_{x}}\\ T_{y}=\frac{a_{8}T_{z}}{f_{y}}\\ T_{x}=\frac{a_{4}T_{z}-\gamma T_{y}}{f_{x}} \end{array}\right.\left\{ \begin{array}{l} r_{5}=\frac{a_{6}T_{z}}{f_{y}}\\ r_{8}=a_{10}T_{z}\\ r_{2}=\frac{a_{2}T_{z}-\gamma r_{5}}{f_{x}}\\ \left(\begin{array}{ccc} r_{3} & r_{6} & r_{9} \end{array}\right)=\left(\begin{array}{ccc} r_{1} & r_{4} & r_{7} \end{array}\right)\times \left(\begin{array}{ccc} r_{2} & r_{5} & r_{8} \end{array}\right) \end{array}\right.$$

So far, all the final parameters’ initial values without distortion coefficients have been solved linearly. The geometry constraints have fully confirmed the orthogonality of the rotation matrix. There is no need to further approximate the rotation matrix’s initial value.

### 4.2. Parameters’ Nonlinear Optimization

Section 4.1 has given the method of parameters’ linear initial values solving. Combining Equations (4)–(8), parameters to be calibrated are summed as follows:(13)$$\left\{ \begin{array}{l} \beta ={\left(\begin{array}{cc} \beta_{x} & \beta_{y} \end{array}\right)}^{T}\\ K_{\mathrm{IACC}}=\left[\begin{array}{ccc} f_{x} & \gamma & 0\\ 0 & f_{y} & 0\\ 0 & 0 & 1 \end{array}\right]\\ P_{c}=(U_{0},V_{0})\\ R_{\mathrm{vector}3\times 1}=Rodrigues\left(\left[\begin{array}{ccc} r_{1} & r_{2} & r_{3}\\ r_{4} & r_{5} & r_{6}\\ r_{7} & r_{8} & r_{9} \end{array}\right]\right)\\ T={\left(\begin{array}{ccc} T_{x} & T_{y} & T_{z} \end{array}\right)}^{T}\\ D={\left(\begin{array}{ccccc} k_{1} & k_{2} & p_{1} & p_{2} & k_{3} \end{array}\right)}^{T} \end{array}\right.$$

With the improvement of the lenses’ manufacturing process, the distortion of today’s non-wide-angle lenses of cameras is very small. The initial guess of ***D*** can be simply set to ***0***. And the initial guess of (*U_0_*, *V_0_*) can be set to the centre of the collected images. As one of the non-coplanar calibration methods’ advantages, there is only one set of intrinsic and extrinsic parameters for all collected images. Equation (13) illustrates the Rodrigues transformation, which supplies the interconversion of rotation ***R***_3×3_ matrix and ***R***_3×1_ vector.

The minimum error squared sum objective function for pixel coordinates can be established:(14)$$\begin{array}{l} \left\{ \begin{array}{l} f_{NUi}=U_{i}-{\hat{U}}_{di}(K_{\mathrm{IACC}},P_{c},R_{\mathrm{vector}3\times 1},T_{\mathrm{vector}3\times 1},D,\beta_{x},\beta_{y})\\ f_{NVi}=V_{i}-{\hat{V}}_{di}(K_{\mathrm{IACC}},P_{c},R_{\mathrm{vector}3\times 1},T_{\mathrm{vector}3\times 1},D,\beta_{x},\beta_{y}) \end{array}\right.\\ I(K_{\mathrm{IACC}},\;P_{c}\;,R_{\mathrm{vector}3\times 1},\;T_{\mathrm{vector}3\times 1},\;D,\;\beta_{x},\;\beta_{y})=\sum_{i=1}^{N}f_{NUi}{}^{2}+f_{NVi}{}^{2}=\min \end{array}$$

In which, $\left({\hat{U}}_{di},{\hat{V}}_{di}\right)$ is the target’s feature point projection from affine space coordinate system *O_p_-X_p_Y_p_Z_p_* to the camera image sensor’s two-dimensional pixel coordinate system *O_R_-UV*. (*U_i_*, *V_i_*) is the corresponding feature point’s coordinate extracted from the image. 

This paper uses the Levenberg-Marquardt algorithm to solve this nonlinear minimization problem of the monocular camera system as shown in Equation (14). Experiments verify that our method can well converge to provide optimum values when the initial guess of the parameters is well estimated. 

### 4.3. Binocular Camera System Calibration Method

A binocular camera system can be seen as two related monocular cameras. Thus, one of the strategies to calibrate a binocular camera system is the combination of two related monocular camera calibrations. As shown in Figure 3d, when using the binocular camera system to take some mono-view non-coplanar 2D targets’ images in a common viewing field, the collected images can then be used to implement binocular camera system calibration. The same as the monocular calibration mentioned before, there is no need for extra equipment making a sliding direction vertical to a target’s plane by our novel method.

At first, for each single camera, repeat the process from Equations (10)–(12) to calculate the initial value of parameters. The parameters to be calibrated in binocular systems can be summed as:(15)$$\mathrm{Left}\;\mathrm{Camera}:\left\{ \begin{array}{l} \beta ={\left(\begin{array}{cc} \beta_{x} & \beta_{y} \end{array}\right)}^{T}\\ K_{a-\mathrm{IACC}}=\left[\begin{array}{ccc} f_{ax} & \gamma_{a} & 0\\ 0 & f_{ay} & 0\\ 0 & 0 & 1 \end{array}\right]\\ P_{a-c}=(U_{a0},V_{a0})\\ R_{a-\mathrm{vector}3\times 1}=Rodrigues\left(\left[\begin{array}{ccc} r_{a1} & r_{a2} & r_{a3}\\ r_{a4} & r_{a5} & r_{a6}\\ r_{a7} & r_{a8} & r_{a9} \end{array}\right]\right)\\ T_{a}={\left(\begin{array}{ccc} T_{ax} & T_{ay} & T_{az} \end{array}\right)}^{T}\\ D_{a}={\left(\begin{array}{ccccc} k_{a1} & k_{a2} & p_{a1} & p_{a2} & k_{a3} \end{array}\right)}^{T} \end{array}\right.\;\mathrm{Right}\;\mathrm{Camera}:\;\left\{ \begin{array}{l} \beta ={\left(\begin{array}{cc} \beta_{x} & \beta_{y} \end{array}\right)}^{T}\\ K_{b-\mathrm{IACC}}=\left[\begin{array}{ccc} f_{bx} & \gamma_{b} & 0\\ 0 & f_{by} & 0\\ 0 & 0 & 1 \end{array}\right]\\ P_{b-c}=(U_{b0},V_{b0})\\ R_{b-\mathrm{vector}3\times 1}=Rodrigues\left(\left[\begin{array}{ccc} r_{b1} & r_{b2} & r_{b3}\\ r_{b4} & r_{b5} & r_{b6}\\ r_{b7} & r_{b8} & r_{b9} \end{array}\right]\right)\\ T_{b}={\left(\begin{array}{ccc} T_{bx} & T_{by} & T_{bz} \end{array}\right)}^{T}\\ D_{b}={\left(\begin{array}{ccccc} k_{b1} & k_{b2} & p_{b1} & p_{b2} & k_{b3} \end{array}\right)}^{T} \end{array}\right.$$

The careful reader may notice that we have repeatedly calculated $\beta_{x}$, $\beta_{y}$, and $\beta_{z}$ separately in two single cameras’ initial value solving processes. Theoretically, $\beta_{x}$, $\beta_{y}$, and $\beta_{z}$ should remain the same for each single camera in a binocular system.

Our solution for this problem is to use either solution of ($\beta_{x}$,$\beta_{y}$) as the initial value of the binocular system. Then, take the other initial values of final parameters as shown in Equation (15), along with ($\beta_{x}$,$\beta_{y}$), into the nonlinear optimization procedure. 

Then, the minimum error squared sum objective function for pixel coordinates can be established:(16)$$\begin{array}{l} \left\{ \begin{array}{l} f_{NUai}=U_{ai}-{\hat{U}}_{adi}(K_{a-\mathrm{IACC}},{\;P}_{a-C}{,\;R}_{a-\mathrm{vector}3\times 1},{\;T}_{a-\mathrm{vector}3\times 1},{\;D}_{a},\beta_{x},\;\beta_{y})\\ f_{NVai}=V_{ai}-{\hat{V}}_{adi}(K_{a-\mathrm{IACC}},{\;P}_{a-C}{,\;R}_{a-\mathrm{vector}3\times 1},{\;T}_{a-\mathrm{vector}3\times 1},{\;D}_{a},\beta_{x},\;\beta_{y})\\ f_{NUbi}=U_{bi}-{\hat{U}}_{bdi}(K_{b-\mathrm{IACC}},{\;P}_{b-C}{,\;R}_{b-\mathrm{vector}3\times 1},{\;T}_{b-\mathrm{vector}3\times 1},{\;D}_{b},\;\beta_{x},\;\beta_{y})\\ f_{NVbi}=V_{bi}-{\hat{V}}_{bdi}(K_{b-\mathrm{IACC}},{\;P}_{b-C}{,\;R}_{b-\mathrm{vector}3\times 1},{\;T}_{b-\mathrm{vector}3\times 1},{\;D}_{b},\;\beta_{x},\;\beta_{y}) \end{array}\right.\\ I\left(K_{a-\mathrm{IACC}},{\;P}_{a-C}{,\;R}_{a-\mathrm{vector}3\times 1},{\;T}_{a-\mathrm{vector}3\times 1},{\;D}_{a},{\;K}_{b-\mathrm{IACC}},{\;P}_{b-C}{,\;R}_{b-\mathrm{vector}3\times 1},{\;T}_{b-\mathrm{vector}3\times 1},{\;D}_{b},\;\beta_{x},\;\beta_{y}\right)\\ =\sum_{i=1}^{N}f_{NUai}{}^{2}+f_{NVai}{}^{2}+f_{NUbi}{}^{2}+f_{NVbi}{}^{2}=\min \end{array}$$

In which, (${\hat{U}}_{adi}$,${\hat{V}}_{adi}$) and (${\hat{U}}_{bdi}$,${\hat{V}}_{bdi}$) are the target’s feature point projection separately from affine space coordinate system *O_p_-X_p_Y_p_Z_p_* to Left and Right camera image sensor’s two-dimensional pixel coordinate system *O_Ra_-U_a_V_a_* and *O_Rb_-U_b_V_b_*. (*U_ai_*, *V_ai_*) and (*U_bi_*, *V_bi_*) are the corresponding feature points’ coordinates extracted from images of different cameras. 

While familiar with monocular calibration, this paper uses the Levenberg-Marquardt algorithm to solve the nonlinear minimization problem of the binocular camera system as shown in Equation (16). Experiments verify that our method can well converge to the optimum value when the initial guess of parameters is well estimated. It is worth mentioning that the present IACC calibration method has good universality and stability for conventional binocular camera systems.

### 4.4. Novel Simple Circle Feature Points Extraction Algorithm with High Accuracy and Stability Based on Local-ROI-OTSU and Radial Section Scanning Method

Datta et al. [31] and other scholars have verified that using a circles pattern can obtain better calibration accuracy than a chessboard pattern in most instances. And refinement based on iterative method [31], inverse rendering [32], or image rectify [26], etc., can really help improve the extraction accuracy of features. However, the above strategies are built on prior knowledge of the special information between camera and targets. The implementation of the above strategies is not simple. Also, they do not consider an algorithms’ stability to illumination, rotations, etc.

This paper proposes a simple circle feature points extraction algorithm with high accuracy and stability based on Local-ROI-OTSU and radial section scanning method. The introduction and deduction are in Appendix B, and the improvements in accuracy and stability brought with our novel algorithm are verified in Section 5.2, Section 5.3, Section 5.4 and Section 5.5.

## 5. Experiments Results and Discussion

Several experiments are set to test the performance of the proposed methods in this paper. The model of camera used in this paper is Basler acA1300-60gm, for which resolution is 1280 × 1024, for which pixel size is 5.3 µm × 5.3 µm, and for which there are matching 12 mm lenses. Three classical and typical calibration methods—Tsai’s method [5], Zhang’s method [9], and ACC method [8]—are used as the contrast methods in the experiments.

Firstly, carry out simulation experiments in Section 5.1 to analyze the performance of our calibration method with respect to the noise level, the number of planes, and different yaw angles. Stability simulations of the proposed novel circle feature points extraction algorithm are carried out in Section 5.2 to evaluate the stability with respect to illumination and viewing angle changes.

Then, carry out the real calibration experiments for the monocular camera in Section 5.3 and Section 5.4. Calibrate the intrinsic and extrinsic parameters of the two cameras respectively by the proposed IACC method and the three contrast methods. Evaluate the accuracy of the resulting parameters from multiple aspects. 

Further, carry out the real calibration experiments for the binocular camera system in Section 5.5. Zhang’s method, ACC method, and the proposed IACC method are separately used to calibrate the binocular camera system’s unknown parameters. Then, take 3D reconstruction experiments with calibrated parameters for discrete feature points to test the actual measurement accuracy. Compare measured distance between points with the actual values. Experiments also verify the deduction in Section 2.3 about the local spatial optimality of calibration parameters.

In Section 5.6, use stereo-DIC method with calibrated binocular systems to carry out full-field stereo measurements based on 3D reconstruction. The results show the feasibility of applying our IACC calibration method for both discrete feature points and surface full-field measurements. 

To ensure objectivity, we use the same calibration targets of different patterns to set experiments comparing different calibration methods. The patterns processing accuracy of the targets is 1 μm. Different feature points extraction algorithms are used to obtain points’ sub-pixel coordinates and make contrast experiments. Zolix KA50 motorized linear stage with MC600 controller is used to generate displacements of fixed direction for non-coplanar methods. Attocube IDS3010 laser interferometer is used to monitor the 1D out-of-plane shifts of targets’ plane.

### 5.1. Performance Simulations of Proposed IACC Calibration Method with Respect to the Noise Level, the Number of Calibration Images, and the Rotation Angle of Targets’ Plane

The simulated camera has the following properties: *f_x_* = 2255.0, *f_y_* = 2254.8, *γ* = 0.05, (*U*_0_, *V*_0_) = (640, 512), (*k*_1_, *k*_2_) = (−0.005, 0.005), and (*p*_1_, *p*_2_) = (0.001, 0.001). The target’s plane contains 11 × 8 = 88 feature points, and the distance between nearby feature points is 10 mm. The image solution is 1280 × 1024. 

**Performance with respect to the noise level.** In this experiment, we use 10 planes in mono-viewing multiplane position to simulate the monocular camera calibration. The extrinsic parameters are set as follows: (*R*_*v*1_, *R*_*v*2_, *R_v3_*) = (0, 0, 0), (*T_x_*, *T_y_*, *T_z_*) = (−64.0, −51.2, 225.0), and (*β_x_*, *β_y_*) = (0.087, 0.000). Gaussian noise with 0 mean and *σ* standard deviation is added to the projected image points. We vary the noise level from 0.1 pixels to 2.0 pixels. For each noise level, we perform 100 independent trials, and the results shown are the average. As we can see from Figure 6b, the relative errors in *f_x_* and *f_y_* are less than 0.4%. For the most simulated noise level, they are less than 0.3%. Other intrinsic parameters, i.e., *γ*, *U_0_* and *V_0_* show similar properties as *f_x_* and *f_y_*. Just as shown in Figure 6, they have very good accuracy and stability. The intrinsic parameters’ average calibration results are not as sensitive to the noise level as Zhang’s method. Reference [9] mentioned that the simulated intrinsic parameters’ errors by Zhang’s method increase linearly with the noise level. For *σ* = 0.5, the errors in *f_x_* and *f_y_* with Zhang’s method are less than 0.3%. Thus, our method shows better stability than Zhang’s when the noise level is less than 2.0 pixels. For the extrinsic parameters, their errors also remain within a reasonable range. When the noise level is lower than 1.4 pixels, the rotation angle’s error is less than 0.02°, as well as the max error of (*T_x_*, *T_y_*, *T_z_*) is less than 3 mm. The error of (*β_x_*, *β_y_*) is less than 0.015, which means the translation direction’s calibration error is less than 0.015°. The distortion coefficients’ error remains low when the noise level is lower than 2 pixels, especially when it is lower than 1.4 pixels. It is worth mentioning that the reprojection error of the proposed calibration method could well converge on the ground truth error value we set before when the noise level is less than 2.0 pixels, as shown in Figure 6a.

The standard deviation of the parameters’ calibration result is used to characterize the uncertainty of the results. As shown in Figure 6e, the uncertainty of *f_x_* and *f_y_* keeps increasing with the rising noise level. Other parameters’ uncertainty does not show in the figure which has similar characteristics with *f_x_* and *f_y_*. Thus, the noise level of images has a directly negative effect on the uncertainty of the proposed calibration method, which should be noted in practical applications.

**Performance with respect to the number of planes.** In this experiment, the simulated camera has the same intrinsic and extrinsic parameters as the experiment with respect to the noise level. We vary the number of planes in mono-viewing multiplane position from 2 to 20. For each number, we perform 100 independent trials. Independent Gaussian noise with mean 0 and standard deviation 0.5 pixels is conducted in the trials. The results are the average as shown in Figure 7. We can learn from Figure 7b that average relative errors of *f_x_* and *f_y_* decrease significantly when the number of planes increase from 2 to 3. Then, they become quite stable, and the relative error remains lower than 0.3%. The other intrinsic and extrinsic parameters’ errors show similar properties as *f_x_* and *f_y_*. The absolute errors of main distortion coefficients *k*_1_ and *p*_1_ stay close to 0, which shows favourable stability with the number of planes increasing. The errors of high-order radial distortion coefficient *k*_2_ seem like changing more dramatically. In numerical terms, the error is still small, and has little effect on the results. 

The data of the reprojection error shown in Figure 7a illustrates that the proposed calibration method could well converge on the ground truth error value. And the number of planes has little effect on the reprojection error. Further, the uncertainty of *f_x_* and *f_y_* shown in Figure 7e decrease significantly when the number of planes increase from 2 to 7, and then decrease more slightly as the number increases from 7 to 20. 

**Performance with respect to the rotation angle of targets’ plane.** In this experiment, the displacement direction of the calibration target remains parallel to the optical axis of the camera. To examine the influence of the orientation of the target’s plane with respect to the imaging plane, we firstly set the target’s plane parallel to the imaging plane. The target’s plane is then rotated around the *Y_w_*-axis with angle *θ*. The angle *θ* varies from 10° to 50°. From the θ could we obtain the *R_vec(Rv1*, *Rv2*, *Rv3*) = (0, −*θ*(rad), 0), (*β_x_*, *β_y_*, *β_z_*) = (sin(*θ*), 0, cos(*θ*)). The other extrinsic parameters and intrinsic parameters remain the same as the above two experiments. Then, these parameters are used to generate simulation datasets. Independent Gaussian noise with mean 0 and standard deviation 0.5 pixels is added to the projected points. Ten images of simulated feature points-pairs are used to calibrate the camera with different angle *θ*. We repeated this progress 100 times and computed average errors. The results are shown in Figure 8. The data in Figure 8b,d illustrate that the rotation angle has little effect on *f_x_*, *f_y_*, *U*_0_, and *V*_0_ when *θ* increases from 0° to 50°. When *θ* is increasing larger than 40°, the relative error of *f_x_* grows faster. When *θ* increases to 50°, the relative error of *f_x_* increases to around 0.3%, along with the relative error of *f_y_* which is still less than 0.1%. The rotation angle has a relatively large effective on *γ*, especially when *θ* is increasing larger than 20°. Even if the value of *γ* increases to six, it means the angle between the image sensor’s two axes is 89.847° in our simulated camera. The result is very close to 90°, which can be accepted in real situations. As for the extrinsic parameters, *T_x_* and *T_y_* seem relatively more sensitive to the rotation angle than *T_z_*. This can be explained by the simulated rotation direction. Then, the rotation vector’s simulated value is quite close to the ground truth value, and the error of the rotation vector is less than 0.1° for most simulated rotation angles. The distortion coefficients’ errors are low, which shows favourable stability with the increasing rotation angle.

The data of the reprojection error shown in Figure 8a illustrates that the proposed calibration method could well converge on the ground truth error value. And the rotation angle has little effect on the reprojection error. Further, the uncertainty of *f_x_* and *f_y_* as shown in Figure 8e increase relatively significantly when the angles increase from 0° to 50°, and the uncertainty of *f_x_* increases more distinctly than the uncertainty of *f_y_*. This can be explained by the simulated rotation direction. Other parameters’ uncertainty does not show in the figure but has similar characteristics with *f_x_* and *f_y_*. Obviously, the increasing rotation angle may bring in more uncertainty of the calibration parameters. 

The experiments in Section 5.1 can summarize some useful conclusions:The proposed IACC calibration method can fit different levels of noise in images. From the simulation experiment result, our method shows better accuracy and stability than Zhang’s method. However, the increasing noise level will bring in more uncertainty of the calibrated parameters. Thus, it is necessary enhance the certainty of the parameters by reducing the noise level of the feature points’ coordinates.The more images used in the calibration, the less uncertainty the parameters will have. Note that in practice, taking more images means we need more displacement data of 2D targets, which may bring in new uncertainty. Thus, combining with our simulation experiments, the suggested number of images is around 10.The proposed calibration method can fit 2D targets’ plane at different angles with the image plane. Compared with the simulation data in [8], our improved method shows better accuracy and stability than the ACC method with respect to the rotation angle. However, increasing the angle may bring in the difficulty of extracting feature points precisely and the uncertainty of calibration parameters. Thus, try to avoid taking images from a large angle, and experience and data indicate that an angle of less than 45° is suggested.

### 5.2. Stability Simulations of the Proposed Novel Circle Feature Points Extraction Algorithm

Illumination conditions are very important to visual measurements. The edges of image features may occur with 1–2 Pixels offsets while the illumination intensity has a 10~20% change. In actual measurements, it is hard to put forward a uniform standard to evaluate the illumination’s sufficiency and suitability. Illumination conditions are often set according to the experience of the operators. Thus, the stability of feature extraction algorithms with respect to illumination change is quite important for high-precision measurements.

Planar targets of a symmetric circle pattern with a back light source are chosen to be the measurement object. The illumination intensity of the back light source is constant, and engineering parts are used to keep the light source and planar target fixed. For simulating the scenes from insufficient illumination to sufficient illumination, we vary the exposure value of the camera from 600 to 3100 and take images from the front of the target. Then, take the image of 3100 exposure value as the reference image. Separately use present novel circle feature points extraction algorithm in Appendix B and OpenCV’s findCirclesGrid to extract the circles’ centers pixel information. RMS errors in Pixels between the test images and the reference image are used to evaluate the stability of the above two algorithms with respect to illumination changes. The result is shown in Figure 9a. Further test the stability of these two algorithms at different viewing angles. We hold the camera still and separately rotate the target around the central axis with 20° and 45°. Repeat the above procedures to test the stability of the two algorithms at different viewing angles. The test results are separately shown in Figure 9b,c.

The results in Figure 9 clearly show the better performance our novel circle feature points extraction algorithm has with respect to illumination and viewing angles changes than OpenCV’s traditional algorithm findCirclesGrid. Simulations at different angles and illumination levels verify that our novel algorithm can stably extract symmetric circles’ features at different viewing angles and illumination conditions. 

### 5.3. Real Monocular Camera Calibration Experiments

Planar chessboard pattern and symmetric circle pattern are chosen to be the calibration targets’ patterns. First, monocular camera calibration experiments are performed. The same 2D calibration targets are used to perform experiments. Some machined parts are used to fix the targets, stage, and cameras. The information of the calibration targets’ pattern is shown in Table 6.

Use stage and targets to generate a virtual 3D points array. Machine parts are used to keep the sliding direction approximately vertical to the targets’ plane. Take images of both patterns in different positions. Feature point extract functions findChessboardCorners and findCirclesGrid from OpenCV 3.3.0 are used in this section to extract the corresponding image pixel coordinates. 

First, 11 × 8 chessboard pattern is used to perform the monocular camera calibration experiments, whereby 1760 point-pairs from 20 images are used to generate the datasets in Table 7. 

Then, use the above datasets to calibrate two monocular cameras separately by the mentioned four methods in Table 7. The reprojection RMS errors in pixels and the errors in world system between detected feature points and projected ones (also called as reprojection error, but the unit is mm) are used to evaluate the accuracy of these four methods. Table 8 and Table 9 show the calibration results of two different cameras by mentioning four methods with chessboard datasets in Table 7. 

The results in Table 8 and Table 9 show that the present IACC method with the traditional chessboard pattern has better accuracy than Tsai’s method, ACC method, and Zhang’s method. 

The other target of 9 × 7 symmetric circle pattern is used to perform the monocular camera calibration experiments with the above four methods. In all, 630 feature points-pairs data from 10 images is used to calibrate the unknown parameters. The data sets of the symmetric circle pattern in this section are made by using findCirclesGrid from OpenCV 3.3.0. The calibration results are shown in Table 10 and Table 11. The results in Table 10 and Table 11 show that the present IACC method with the traditional symmetric circle pattern also has better accuracy than Tsai’s method, ACC method, and Zhang’s method.

Clearly, Table 8, Table 9, Table 10 and Table 11 also reflect how different calibration methods using symmetric circle pattern with OpenCV’s findCirclesGrid has better performance than using chessboard pattern with findChessboardCorners. The simulation results in Section 5.1 verify that the reprojection error may be convergent to the added noise of feature points. Certainly, there are other noises in actual images. But the accuracy of the feature points’ extraction algorithm plays a major part in added noise. Thus, the reprojection error could reflect the accuracy of the points’ extraction algorithm. From Table 9 and Table 10’s data, the accuracy of findCirclesGrid can achieve close to 0.02 pixels, and the accuracy of findChessboardCorners can only come close to 0.06 pixels in our real experiments. 

Comparing the calibration results for different cameras in Table 10 and Table 11, we can also notice that the improvements brought by IACC for Single_R camera is relatively lower than for Single_L camera. After the examination, we found that there are some imperceptible stains on the surface of Single_R camera’s imaging sensor, which may affect the quality of the calibration images. The feature extraction algorithm findCirclesGrid is easily affected by these stains because findCirclesGrid is a gray-centroid-based blob detect algorithm. Clearly, alongside the calibration model, the accuracy of feature extraction will more directly affect the accuracy of calibration. Thus, the stability of algorithms for different measurement environments is important, and our new algorithm in Appendix B has better performance than findCirclesGrid. This has been verified in simulations in Section 5.2 and in real calibration experiments in Section 5.4.

As for distortion coefficients, Tsai’s method assumes tangential distortion can be ignored to satisfy RAC constraint, so Tsai’s method cannot calibrate the tangential distortion. The present IACC method, ACC method, and Zhang’s method can calibrate the radial and tangential distortion coefficients through nonlinear optimization. It needs to be explained that different coefficients definition expression in [8] and in this paper caused difference in values in Table 8, Table 9, Table 10 and Table 11. Both coefficients definition expression can meet the physical model and can reflect the distortion level. For convenience, the coefficients definition expression in this paper is in accordance with Zhang’s method [9].

### 5.4. Performance of Present New Algorithm in Appendix B for Improving Calibration Accuracy

We further apply the present new algorithm in Appendix B to generate feature point-pairs data sets with the same calibration images in Table 5 and Table 6. The present IACC calibration method and Zhang’s method are chosen to verify that our novel circle feature points extraction algorithm can make improvements of calibration accuracy for both non-coplanar calibration methods and coplanar calibration methods.

The same as Table 10 and Table 11, 630 feature points-pairs data from 10 images are used to calibrate the unknown parameters. Next, Δθ is set to 1° and the searching step length in radial direction is set to 0.1 pixel. The calibration results are shown in Table 12 and Table 13.

The results data in Table 12 and Table 13 clearly verified that our novel circle feature points extraction algorithm effectively improves the calibration accuracy of both the coplanar method and the non-coplanar method, represented by Zhang’s method and the present IACC method. From previous deduction that the reprojection error may be convergent to the accuracy of feature points extraction algorithm with the present method, we could further reckon that the present new algorithm in Appendix B has better accuracy than findCirclesGrid from OpenCV 3.3.0. The accuracy of our algorithm can reach within 0.02 Pixels in actual application.

As we mentioned before, the implementation of coplanar calibration methods needs to take images of 2D targets in multi-viewing multiplane position as shown in Figure 3. The data in Table 12 and Table 13 also illustrates the stability of the proposed novel circle feature points extraction algorithm for the rotation of planar targets. As shown in Figure 10, our algorithm and strategy to sort the key points can fit the situations when angular deflection between planar targets and imaging sensor remains at a relatively rational level. Usually, this angle should be below 45° to keep the accuracy of calibration results. 

Combining the simulations in Section 5.2, the present new algorithm in Appendix B has better extraction accuracy and stability than the traditional extraction algorithm findCirclesGrid from OpenCV 3.3.0, whereby our novel algorithm can effectively improve the calibration accuracy for both non-coplanar methods and coplanar methods. 

### 5.5. Real Binocular Camera System Calibration and 3D Reconstruction Experiments for Discrete Feature Points

We set experiments to test the performance of the proposed binocular camera system calibration method. As a supplement, a 3D reconstruction experiment for discrete feature points is set to evaluate the actual measurement accuracy of the binocular measurement system calibrated by the proposed method. 

Firstly, use the binocular camera system to acquire a set of images to be the test images. Take part of the test images to be the parameter calibration images and the other part to be the accuracy test images. Then, measure the distance between different feature points on the target surface and compare the measured distance data with actual values.

Similar to the monocular calibration process, we fixed a one-dimensional displacement stage with the planar target of high precision on the optical platform, moved the planar target in a fixed direction, and used the laser interferometer to measure the moving shifts of the calibration target in this direction. Then, we established the affine space coordinate sequence of the calibration target. The intrinsic and extrinsic parameters of the binocular system can be calibrated through the target’s image sequence captured by the two cameras simultaneously.

The binocular system is shown as Figure 11. The machine part is designed to fix two cameras. The designed horizontal baseline between two cameras is 156 mm, and the included angle between the camera optical axis and the baseline is about 75°. Five out of ten image-pairs of the symmetric circle pattern acquired at mono-viewing positions are taken to calibrate the parameters of the binocular system. The other five image-pairs are retained for the distance measurement experiment.

After the calculation process mentioned in Section 4.3, the intrinsic and extrinsic parameters of the binocular camera system have been calibrated simultaneously. The extrinsic parameters of the two cameras in the system can be expressed in more universal format by Equation (17):(17)$$\{\begin{array}{c} R_{a3\times 3}=Rodrigues(R_{a-\mathrm{vector}3\times 1})\\ \begin{array}{l} R_{b3\times 3}=Rodrigues(R_{b-\mathrm{vector}3\times 1})\\ R_{a-b3\times 3}=R_{b-3\times 3}\cdot R_{a-3\times 3}{}^{-1}\\ T_{a-b}=T_{b}-R_{a-b3\times 3}\cdot T_{a} \end{array} \end{array}$$

The calibration results of the proposed method are shown in the bottom part of Table 14.

It is worth noting that reference [8]’s binocular system calibration method (ACC method) can only calibrate extrinsic parameters with the prerequisite known monocular camera’s intrinsic parameters. The ACC binocular system calibration method cannot calibrate the intrinsic and extrinsic parameters simultaneously. And the ACC binocular system calibration method introduced a penalty function to ensure the orthogonality of the rotation matrix. This means that the penalty factors should be adjusted along with the changes in the binocular cameras’ structure, calibration targets’ pattern, monocular calibration accuracy, etc. Above all, in actual applications, the loss of universality makes the ACC method too cumbersome to carry out the binocular camera systems’ calibration. The calibrated extrinsic parameters of the binocular system and the prerequisite intrinsic parameters with the ACC method are shown in the top-left of Table 14. 

Similar with the ACC method, the complicated procedure to adjusting the targets’ sliding direction of Tsai’s non-coplanar method also make it inefficient to calibrate either monocular or binocular camera systems with high accuracy. Considering how the accuracy of the previous monocular calibration with Tsai’s method under current conditions is much more worthwhile than the other three methods, Tsai’s method is not considered to be the contrast method in this section. 

The same as the present IACC method, Zhang’s method can calibrate the intrinsic and extrinsic parameters of the binocular simultaneously. Zhang’s binocular calibration method which contained in OpenCV 3.3.0 function stereoCalibrate is chosen as a comparative method. Without loss of generality, five out of ten image-pairs of the symmetric circle pattern from multi-viewing positions are taken to calibrate the parameters of the binocular system with Zhang’s method. The other five image-pairs are retained for the distance measurement experiment. The calibration results of Zhang’s method are shown in the top-right part of Table 14.

The key data reprojection errors in Table 14 reflect how our binocular calibration method has the best calibration accuracy among the mentioned three methods. It is worth mentioning that the feature points of the symmetric circle pattern used in the above experiments are extracted by the present new algorithm in Appendix B for all three binocular calibration methods.

As mentioned in Section 2.4, calibration parameters always present local spatial optimality. Thus, we should not discuss the calibration accuracy of the binocular camera system in isolation ignoring the actual measurement position. This means that we should combine the actual measurement accuracy with calibration accuracy to evaluate the actual accuracy of calibrated parameters. 

Camera-based stereo measurements are built on the 3D reconstruction of features in images acquired by camera-sensing-systems. The 3D reconstruction of image features mainly contains three steps, i.e., features’ extraction, features’ stereo matching, and features’ stereo reconstruction based on triangulation. The accuracy of the camera-based stereo measurements mostly depends on the accuracy of the features’ matching and multi-camera system’s calibration. With precise camera calibration parameters and matching point-pairs’ coordinates, based on the 3D reconstruction algorithm, high-precision measurements can be realized.

Clearances measurements for discrete circular feature points based on 3D reconstruction are set for further testing the accuracy of the calibrated parameters. The clearances to be measured can be divided into two types as shown in Figure 12. The clearances between feature points in high precision calibration planar targets are used as the measurement objects. As shown in Figure 12, a specific image-pair of 9 × 7 symmetric circle pattern corresponds to a planar target at one specific position in a world coordinates system. For one specific position, each target can acquire 56 sets of horizontal clearances and 54 sets of vertical clearances. 

In this paper, the least squares method is used to solve the 3D reconstruction for discrete circular feature points, since this method can directly use the original matched feature point-pairs’ pixel information and calibrated parameters without extra image affine transformation and interpolation. 

Different camera parameters calibrated by the mentioned three methods are used to perform the 3D reconstruction of the discrete feature points with the above retained image-pairs. The error data is shown in Table 15 and Table 16.

Data in Table 14, Table 15 and Table 16 illustrate that the parameters of the binocular camera system calibrated by the present IACC method not only have the best calibration accuracy among three contrast methods, but also can supply the best measurements accuracy for circular feature points at nearby positions of where the calibration images are acquired. Data in Table 15 and Table 16 can also reflect that parameters calibrated by different methods achieve their best measurements accuracy at around the calibration position, but cannot achieve equivalent measurements accuracy away from the calibration position. As we can see from the root mean square error data in Table 15 and Table 16, the binocular system with IACC’s calibration parameters can achieve 2.6 μm measurement accuracy using images taken from the mono-viewing multiplane position, but can only achieve 59.1 μm measurement accuracy using images taken from the multi-viewing multiplane position. Similarly, the binocular system with Zhang’s calibration parameters can achieve 21.7 μm measurement accuracy using images taken from the multi-viewing multiplane position, but can only achieve 52.0 μm measurement accuracy using images taken from the mono-viewing multiplane position. Referring to Figure 13, the mono-viewing position and the multi-viewing position in this experiment are clearly distributed in different depths of the same world coordinates system. These results have verified the local spatial optimality of the calibration parameters mentioned in Section 2.4. 

The parameters either from non-coplanar or coplanar methods obtain the best effect when their calibration positions are close to and cover most of the measurement space. Overall, the present IACC calibration method for binocular camera systems shows prominent advantages in calibration accuracy and measurements accuracy than both Zhang’s method and the ACC method.

The left camera’s coordinates system is chosen to be the world coordinates system, using IACC’s calibration parameters and feature extraction algorithm, wherein the calculated 3D coordinates of the measured feature points’ centers are drawn in Figure 13. Figure 13 can clearly show the difference of the measured targets’ positions in the world coordinates system. 

### 5.6. Full-Field Stereo Measurement Experiments by Stereo-DIC Technologies with the Proposed Calibration Method

In the last few decades, stereo-Digital Image Correlation (stereo-DIC) has been widely accepted as a powerful and versatile tool for non-contact full-field 3D shape and surface deformation measurement in experimental solid mechanics [22,33]. Stereo-DIC relies on the image correlation analysis of image-pairs obtained from a calibrated stereo-vision system. Stereo-DIC is still far from reaching its full potential. This is mainly due to three major challenges that Sutton and associates [34,35] identified as follows: (1) surface patterning; (2) imaging of the structure (i.e., appropriately selecting lens and stereo-angle); (3) calibrating the stereo-DIC measurement system.

Among the various calibration techniques used in the computer vision community, the two traditional methods presented by Zhang [9] and Tsai [5] are still commonly taken as key-methods for stereo-DIC system calibration with 2D and 3D targets, respectively. Research on the calibration methods applied for the stereo-DIC system is still valuable, and we will further testify the feasible applications of the proposed novel calibration methods in full-field stereo measurements.

The calibrated binocular camera system in Section 5.5 and three acrylic hollow cylinders with artificial speckle patterns are used to carry out full-field 3D shape measurements. The speckle patterns are arranged into cylinders’ surfaces by hydro transfer printing.

Two classical stereo-DIC methods, Newton–Raphson (NR) [36] method and inverse-compositional Gauss–Newton (IC-GN) [37] algorithm, are used to carry out stereo matching for the speckle patterns’ subsets region. 

The Newton–Raphson (NR) method has been integrated into an open-source digital image correlation (DIC) tool DICe [38] from Sandia National Laboratories. Its primary capabilities are computing full-field displacements and strains from sequences of digital images and rigid body motion tracking of objects.

A calibrated binocular camera system is used to take three image-pairs of cylinder objects with different radii as shown in Figure 14.

Calibrated parameters from Zhang’s method and the IACC method are separately taken into the DICe to supply the basic parameters of the binocular stereo-DIC system. Every image-pair from single capturing at same moments are used to be both the reference image and the deformed image. Thus, the calculated displacement and deformation of the objects should be zero theoretically. The measurement results separately calculated with Zhang’s parameters and with IACC’s parameters are very close, and the similarity of these results can be reflected by colormaps in Figure 15. Since colormaps can only reflect the rough tendency, if the accuracy of the calibration parameters is close enough, with the same high-precision DIC matching method, one can barely tell the difference between the top and bottom parts of Figure 15.

Figure 15 shows the static measurement results of one cylinder. Figure 15a–c separately demonstrates the measured ROI’s z-coordinates, displacement of x-direction, and normal strain of x-direction with the binocular system calibrated with Paper’s parameters. As we expected, the calculated displacement and deformation of the measured ROI are very close to 0. And the z-coordinates of the measured ROI accord with the actual situation of the measured position. It is worth noting that the static measured displacement data of ROI is quite close to zero and the absolute error value is mostly within 20 picometers. This level of absolute error reflects that both the matched accuracy of DICe and the accuracy of our calibration method remain at elevated levels. We also noted that the region of slightly larger error appeared around the circular ring at the middle of the ROI. This can also meet expectations because there is no speckle distribution inside the ring as we designed. As a testified method by many scholars which can be used in the calibration stereo-DIC, the calibration parameters from Zhang’s method are also used to calculate the same image-pair. The results are shown in Figure 15d–f, which shows similar properties as the results achieved with IACC’s parameters. 

Then, full-field 3D reconstruction experiments for surface ROIs’ matched subsets from three different cylinders are carried out by our self-designed IC-GN-based program. The IC-GN method, first-order shape function, and Bicubic interpolation method are used in our program to complete the correlated subsets’ sub-pixel matching. A seed-point-based neighbor-region-generation calculation path is applied to complete the ROI’s full-field stereo matching calculation. Stereo-rectify of image-pairs based on our calibration parameters has been implemented before the ROI subsets’ correlation matching to reduce the deformation of the corrected subsets caused by the different viewing angles of the left and right cameras. The 3D reconstruction results of the ROIs from different cylinders’ surfaces are shown in Figure 16. The results from our program and from DICe are consistent.

The local ROIs’ points cloud data in Figure 17 are used to achieve cylinder fit by the nonlinear least squares method based on the Levenberg-Marquardt algorithm. The cylinder mathematical model can be illustrated as follows:(18)$$\sqrt{{\left(X_{p}-x_{0}\right)}^{2}+{\left(Y_{p}-y_{0}\right)}^{2}+{\left(Z_{p}-z_{0}\right)}^{2}-\frac{{\left[l\cdot (X_{p}-x_{0})+m\cdot (Y_{p}-y_{0})+n\cdot (Z_{p}-z_{0})\right]}^{2}}{\left(l^{2}+m^{2}+n^{2}\right)}}=r$$

In which $\left(x_{0},y_{0},z_{0}\right)$ illustrates a point at the cylinder’s main axis, (l,m,n) illustrates the direction vector of the main axis, $\left(X_{p},Y_{p},Z_{p}\right)$ illustrates any point on the surface of the cylinder, and *r* is the radius of the fitting cylinder. 

The three cylinders are made of acrylic material, and their design radii are 75 mm, 100 mm, and 125 mm. The material characteristics determine that the machining accuracy is not very high. The local ROIs fitting radius data from Table 17 reflects the local ROIs’ curvature radii information, which are generally consistent with the design values. The curvature radius of Cylinder #3 is closer to the design value, which reflects how stereo measurements with the binocular camera system work better when the targets’ curve surface has less curvature radius. Obviously, fitting results with 3D point clouds separately calculated by Zhang’s parameters and IACC’s parameters are consistent, and the data in Table 17 can only reflect approximate values of cylinder’s radii. Overall, the fitting results with the present IACC’s parameters are a little closer to the design value of cylinders than with Zhang’s parameters. More experiments need to be conducted in the future to check the real full-field measurement accuracy.

As shown in Figure 17, there are some imperceptible residual adhesive films from the speckle pattern’s transfer printing process retained on the surface of cylinder #2. Clearly, the 3D point clouds from our program have contained flawed information. This reflects the good 3D reconstruction accuracy of our ICGN-based program and calibration parameters. The point clouds data of surface flaws may cause the fitting RMSE error of Cylinder #2 in Table 17 to be a little larger than the errors of Cylinder #1 and #3.

Experiments in Section 5.5 and Section 5.6 can summarize some useful conclusions:The best calibration position should cover the potential measurements spatial range.The present IACC binocular calibration method has the best calibration accuracy among the three contrast methods. And the circular discrete feature points’ measurement accuracy by binocular system with Paper’s calibration parameters and feature extraction algorithm can achieve less than 2.6 μm.The present IACC calibration method can be further combined with classical stereo-DIC technologies, e.g., Newton–Raphson (NR) method and ICGN method, to achieve the surface ROIs’ full-field measurements.Static measurement experiments and 3D reconstruction experiments have shown the feasibility of the present IACC method applied in stereo-DIC system calibration. Loading experiments are still needed for quantitative analysis of the improvement of measurement accuracy lifted by the present IACC calibration method. The quantitative analysis and dynamic loading experiments deserve further research.

### 5.7. Analysis of the Calibration Efficiency of Both Monocular and Binocular Camera Systems

Although calibration accuracy is important, the efficiency of calibration methods is also significant. There are two major factors that determine the efficiency of a certain calibration method. One is the efficiency of the algorithm. The other is the efficiency of the calibration implementation, e.g., the process of feature point-pairs’ coordinates acquisition, adjustment of stage’s sliding direction, arrangement to individually assign some factors for a calibration algorithm, etc. Except for the complexity of the algorithm itself, the efficiency of the algorithm is largely determined by the computer hardware and the programming level. Solely measuring the running time of an algorithm to characterize one’s efficiency is sometimes unfair. In this situation, reference [8] has given the idea of analyzing the complexity of the algorithm itself to reflect the efficiency. However, reference [8]’s efficiency evaluation method did not fully consider the extra implementation complexity of the methods. And reference [8] did not propose the calibration efficiency evaluation method for the binocular camera system, either.

This paper gives a more comprehensive calibration efficiency evaluation method for both monocular and binocular camera systems, the deduction and analysis of which are appended in Appendix C.

According to the evaluation method in Appendix C, assume 10 images are used for calibration, and there are 63 features in every image. The numbers of nonlinear iterations are set to 200, and the *C_operation* is set to 1 × 10^8^. The complexity of the mentioned four methods can be quantified as shown in Table 18.

Thus, from the perspective of algorithm efficiency, for the mentioned four monocular calibration methods, Tsai’s method has the best efficiency, followed by ACC, present IACC, and Zhang’s methods. For the mentioned four binocular calibration methods’ algorithm efficiency, Tsai’s method has the best efficiency, followed by present IACC, Zhang’s, and ACC methods.

From the overall efficiency including algorithm and implementation, for the mentioned four monocular calibration methods, Zhang’s method has the best efficiency, followed by present IACC, ACC, and Tsai’s methods. For the mentioned four binocular calibration methods’ overall efficiency, Zhang’s and present IACC methods have similar best efficiency, followed by Tsai’s and ACC methods.

Summarizing the results from Section 5.1, Section 5.2, Section 5.3, Section 5.4, Section 5.5, Section 5.6 and Section 5.7, this gives our suggestions for the choice of the mentioned four calibration methods:Zhang’s method is the easiest to implement, but the calibration accuracy is not the best. Thus, Zhang’s method is the best choice if there are no extreme demands of high-precision calibration and measurements.The present IACC method for monocular and binocular calibration has the best calibration accuracy and moderate implementation complexity. The present IACC method is the preferred choice for high-precision calibration and measurements. Especially when the structure of camera systems and measurements’ position is confirmed in advance. And in some special scenarios when 2D targets are restricted to change postures in multi-viewing positions, the present IACC method would be the best solution for both accuracy and efficiency.The ACC method can supply accurate calibration parameters for monocular camera systems with good efficiency. However, it is not suitable for binocular camera systems for either accuracy or efficiency.Tsai’s method has distinct defects in mathematical model and is inefficient using planar target with non-coplanar mode. Using real stereo targets will reduce the complexity of Tsai’s method. However, it is still not suitable for high-precision calibration and measurements.The Present algorithm in Appendix B can well serve for the mentioned four methods with high-precision and good stability.

## 6. Conclusions

This paper proposes an Improved Affine Coordinate Correction Mathematical (IACC) model which can be well applied for the calibration of both monocular and binocular camera systems. Our novel calibration methods are stable, efficient, and with high precision. Based on Local-ROI-OTSU and gradient-based Edge Radial Section Scanning method, a novel simple extraction algorithm for the symmetric circles pattern is proposed. The proposed novel algorithm can further improve the accuracy and stability of existing calibration methods.

Performance simulations verify that the present IAAC method possesses good accuracy and stability. The present IACC method can fit different levels of image noise (from 0~2 Pixels), different numbers of planes (at least 2), and 2D targets’ planes at different viewing angles (from 0°~50°). The proper number of images (around 10) and viewing angles (less than 45°) could keep the uncertainty of parameters relatively low and have enough calibration accuracy.

Simulation and real performance experiments are set for the proposed simple novel circle feature points extraction algorithm. Simulation verified that the present new algorithm in Appendix B has better stability with respect to illumination and viewing angles change than the traditional algorithm. Real experiments demonstrate that our new algorithm can significantly improve the calibration accuracy for both coplanar and non-coplanar calibration methods. Calibration results reflect that the feature distraction accuracy of our new circle feature points extraction algorithm can be within 0.02 Pixels. It is worth mentioning that our circle feature points extraction algorithm keeps high accuracy and simplicity without any non-linear iteration or complex rectify method.

Real data in Section 5.3, Section 5.4 and Section 5.5 verify that the accuracy of the present IACC method is better than that of Tsai’s, AAC, and Zhang’s calibration methods for both monocular camera systems and binocular camera systems. The calibration accuracy of the present IACC for the binocular system increases by 10 times than with the AAC method, as well as by 40% than with Zhang’s method. The calibration parameters supported by our IACC method can help real stereo vision system’s measurement accuracy reach within 3 μm for discrete feature points, which is remarkably superior to parameters supported by control methods. 

Our novel IACC calibration methods have further applied for binocular-camera-based full-field stereo measurements based on two classical stereo-DIC methods. Static measurements for displacements and deformations show that calibration parameters supported by the IACC method are feasible to apply in stereo-DIC measurements and have similar accuracy with Zhang’s parameters. All-process 3D reconstruction experiments for cylinders’ surfaces reflects the IACC method’s potential application for calibrating curves surface’s high-precision shape, deformation, and strain measurement visual systems. 

At last, a comprehensive calibration efficiency evaluation method for different calibration methods is given in this paper. According to the analysis, the present IAAC method has the best calibration accuracy and moderate implementation complexity. The present IACC method is the preferred choice for high-precision calibration and measurements. 

In future research, we will focus more on improving stereo rectify strategies for full-field stereo measurements based on high-precision calibration parameters. The quantitative analysis and high-speed dynamic loading experiments for stereo-DIC measurements deserves further research. Research on the calibration target’s pattern combining with DIC measurements’ speckle pattern still deserves more concentration. The combining of our calibration methods with the iterative refinement of the control points’ strategy also deserves more work. 

## Figures and Tables

**Figure 1 sensors-23-08466-f001:**
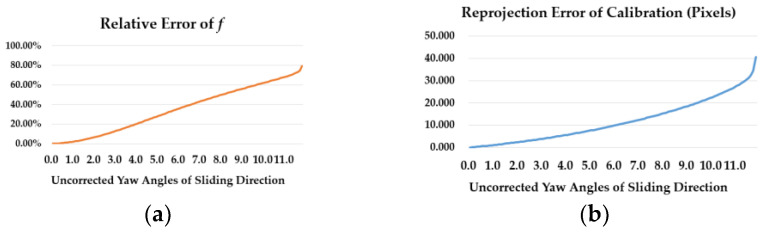
Simulated error data with Tsai’s method. (**a**) shows the relative error of ***f*** changing trend with the increasing uncorrected yaw angle between sliding direction and target’s plane; (**b**) shows the reprojection error changing trend with increasing uncorrected yaw angle between sliding direction and target’s plane.

**Figure 2 sensors-23-08466-f002:**
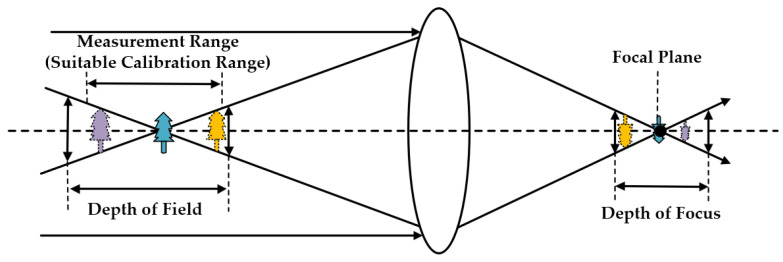
This figure shows the real imaging process and reflects the experience that the measurement accuracy would be better if the calibration feature points closely cover the measurement position.

**Figure 3 sensors-23-08466-f003:**
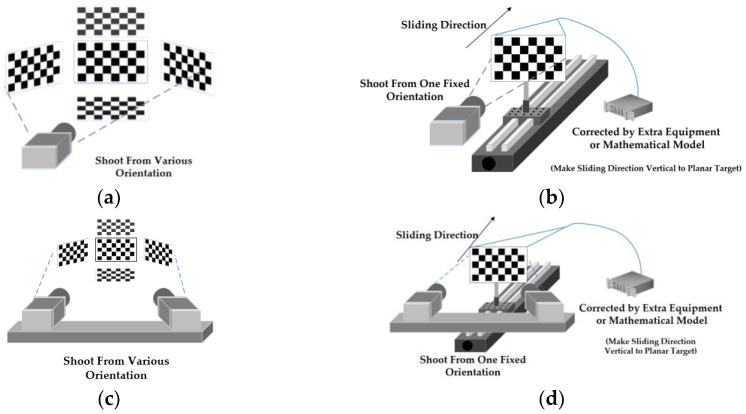
This figure separately shows the implementation processes of coplanar and non-coplanar calibration methods. The multiple panels of this figure are listed as: (**a**) The implementation process of coplanar monocular calibration methods; (**b**) The implementation process of non-coplanar monocular calibration methods; (**c**) The implementation process of coplanar binocular calibration methods; (**d**) The implementation process of non-coplanar binocular calibration methods.

**Figure 4 sensors-23-08466-f004:**
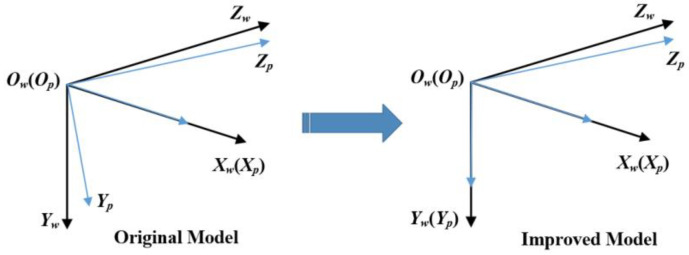
This is a transformation model for mapping a target affine space coordinate system to an orthogonal world coordinate system. This process can be represented as a matrix transformation equation.

**Figure 5 sensors-23-08466-f005:**
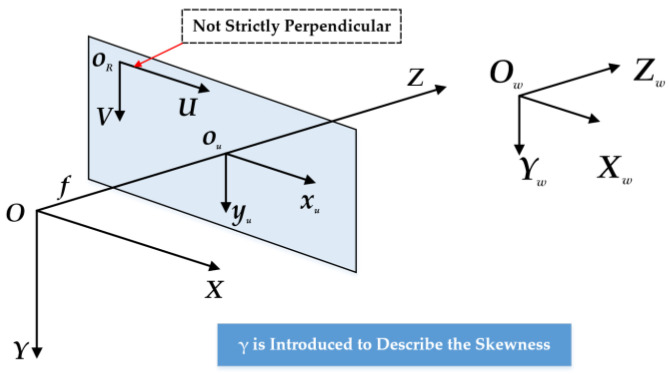
This figure is a camera pinhole imaging model. The transformation among different coordinate system processes can be represented by matrix transformation equations.

**Figure 6 sensors-23-08466-f006:**
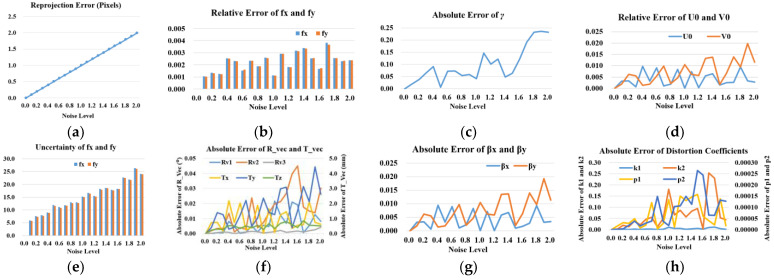
This figure shows the performance of the proposed calibration method with respect to the noise level. The multiple panels of this figure are listed as follows: (**a**) Simulated reprojection error with respect to the noise level; (**b**) Relative error of *f_x_* and *f_y_* with respect to the noise level; (**c**) Absolute error of *γ* with respect to the noise level; (**d**) Relative error of *U_0_* and *V_0_* with respect to the noise level; (**e**) Uncertainty of *f_x_* and *f_y_* with respect to the noise level; (**f**) Absolute error of *R_vec* and *T_vec* with respect to the noise level; (**g**) Absolute error of *β_x_* and *β_y_* with respect to the noise level; (**h**) Absolute error of distortion coefficients with respect to the noise level.

**Figure 7 sensors-23-08466-f007:**
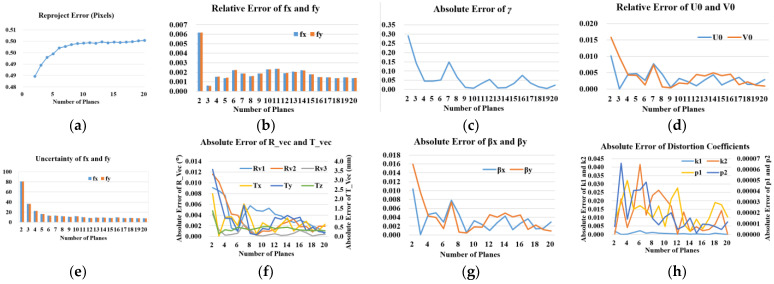
This figure shows the performance of the proposed calibration method with respect to the number of planes. The multiple panels of this figure are listed as follows: (**a**) Simulated reprojection error with respect to the number of planes; (**b**) Relative error of *f_x_* and *f_y_* with respect to the number of planes; (**c**) Absolute error of *γ* with respect to the number of planes; (**d**) Relative error of *U_0_* and *V_0_* with respect to the number of planes; (**e**) Uncertainty of *f_x_* and *f_y_* with respect to the number of planes; (**f**) Absolute error of *R_vec* and *T_vec* with respect to the number of planes; (**g**) Absolute error of *β_x_* and *β_y_* with respect to the number of planes; (**h**) Absolute error of distortion coefficients with respect to the number of planes.

**Figure 8 sensors-23-08466-f008:**
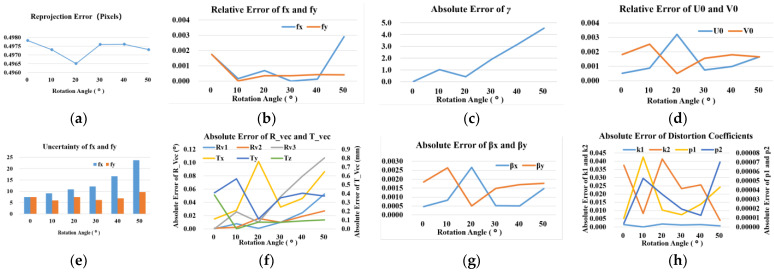
This figure shows the performance of the proposed calibration method with respect to the rotation angle of the targets’ plane. The multiple panels of this figure are listed as follows: (**a**) Simulated reprojection error with respect to the rotation angle of the targets’ plane; (**b**) Relative error of *f_x_* and *f_y_* with respect to the rotation angle of the targets’ plane; (**c**) Absolute error of *γ* with respect to the rotation angle of the targets’ plane; (**d**) Relative error of *U_0_* and *V_0_* with respect to the rotation angle of the targets’ plane; (**e**) Uncertainty of *f_x_* and *f_y_* with respect to the rotation angle of targets’ plane; (**f**) Absolute error of *R_vec* and *T_vec* with respect to the rotation angle of the targets’ plane; (**g**) Absolute error of *β_x_* and *β_y_* with respect to the rotation angle of the targets’ plane; (**h**) Absolute error of distortion coefficients with respect to the rotation angle of the targets’ plane.

**Figure 9 sensors-23-08466-f009:**
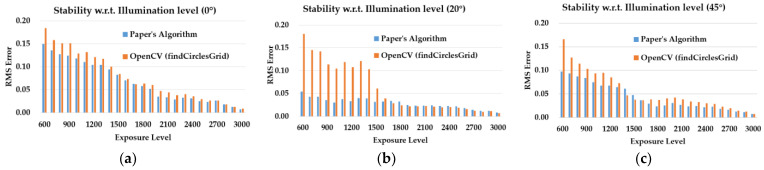
This figure shows the performance simulation results of the two contrast algorithms with respect to illumination and viewing angles changes. The multiple panels of this figure are listed as follows: (**a**) Stability simulations with respect to illumination changes (viewing angle: 0°); (**b**) Stability simulations with respect to illumination changes (viewing angle: 20°); (**c**) Stability simulations with respect to illumination changes (viewing angle: 45°).

**Figure 10 sensors-23-08466-f010:**
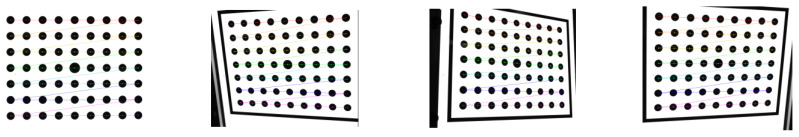
This figure shows the feature searching ability of our novel circle feature points extraction algorithm for fitting different inclination angles of planar targets.

**Figure 11 sensors-23-08466-f011:**
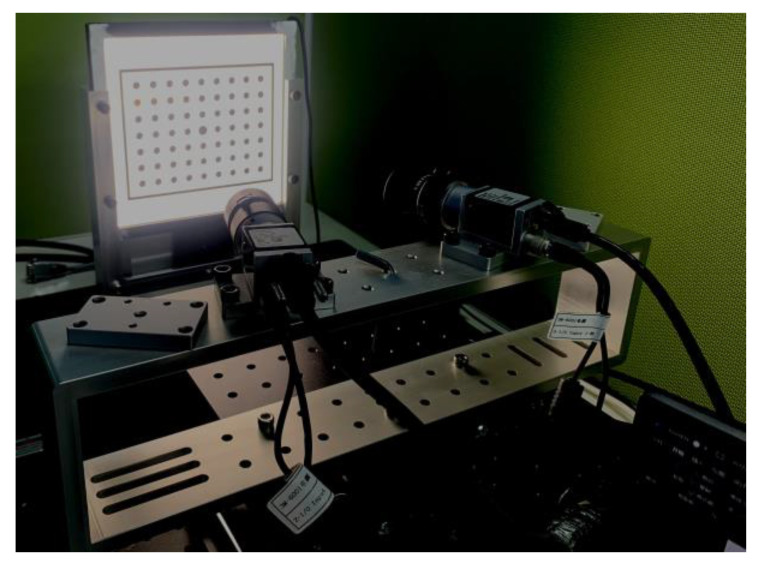
This figure shows the binocular camera system to be calibrated.

**Figure 12 sensors-23-08466-f012:**
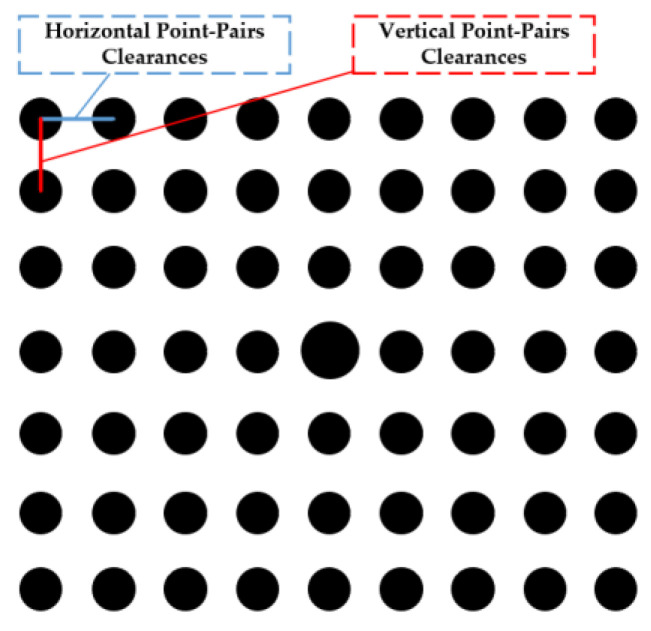
This figure shows the horizontal and vertical point-pairs clearances to be measured in the targets. Each target in a specific position can acquire 56 sets of horizontal clearances and 54 sets of vertical distance clearances.

**Figure 13 sensors-23-08466-f013:**
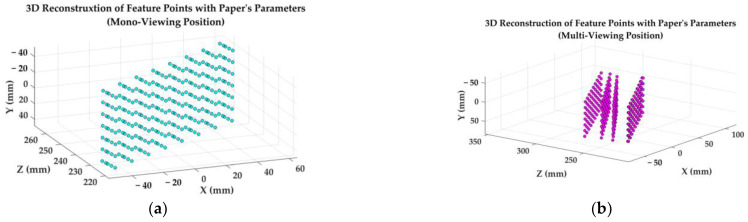
This figure shows a 3D reconstruction of the circular feature points’ centers. Listed as follows: (**a**) 3D reconstruction of feature points with paper’s parameters in mono-viewing muti-plane position; (**b**) 3D reconstruction of feature points with paper’s parameters in multi-viewing muti-plane position.

**Figure 14 sensors-23-08466-f014:**
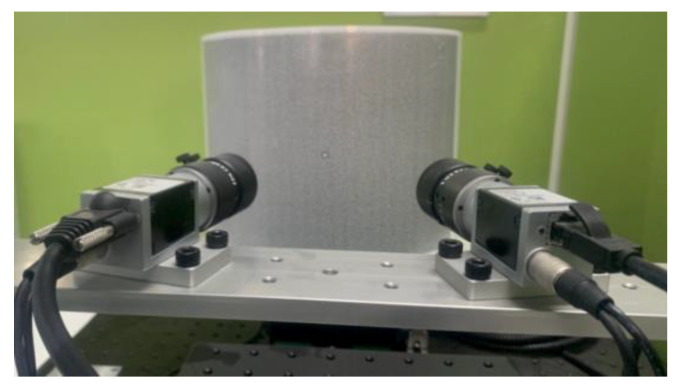
This figure shows the calibrated binocular camera system is used to implement static 3D surface shape reconstruction experiments of cylinder objects with different radii.

**Figure 15 sensors-23-08466-f015:**
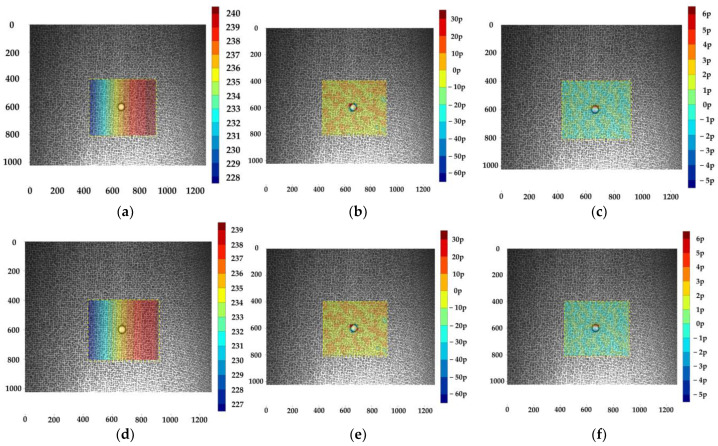
(**a**–**c**) show the static measurement results of the surface’s z-coordinates, displacement of x-direction, and normal strain of x-direction. The results data of (**a**–**c**) comes from the binocular camera system with the present IACC calibration parameters. Correspondingly, (**d**–**f**) show the static measurement results of the surface’s z-coordinates, displacement of x-direction, and normal strain of x-direction. The results data of (**d**–**f**) comes from the binocular camera system with Zhang’s calibration parameters.

**Figure 16 sensors-23-08466-f016:**
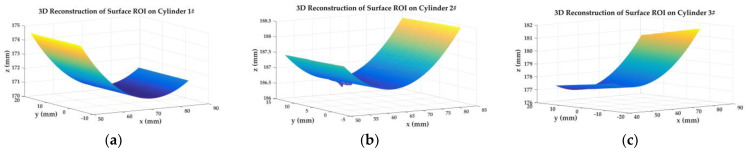
This figure shows 3D reconstructions of the ROIs’ matched subsets from three different cylinders. Listed as follows: (**a**) 3D reconstruction of surface ROI on cylinder #1; (**b**) 3D reconstruction of surface ROI on cylinder #2; (**c**) 3D reconstruction of surface ROI on cylinder #3.

**Figure 17 sensors-23-08466-f017:**
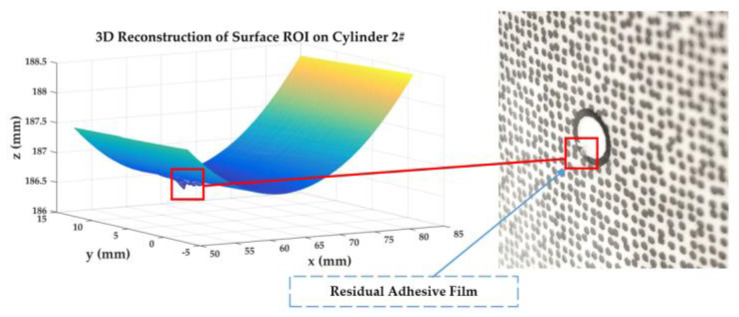
This figure shows the 3D reconstruction of Cylinder #2 with some residual adhesive films retained on the surface of the ROI.

**Table 1 sensors-23-08466-t001:** The mathematical model and algorithm flow of Tsai’s non-coplanar calibration methods.

**Mathematical Model** **(Tsai’s Method)**	$\rho ({\tilde{P}}_{I}-{\tilde{P}}_{C})=K_{\mathrm{Tsai}}\left[\begin{array}{cc} R_{3\times 3} & T_{3\times 1} \end{array}\right]{\tilde{P}}_{W}$	$K_{\mathrm{Tsai}}=\left[\begin{array}{ccc} f\cdot s_{x} & 0 & 0\\ 0 & f & 0\\ 0 & 0 & 1 \end{array}\right]$	$D=(\begin{array}{ccc} k_{1} & k_{2} & k_{3} \end{array})$
			Distortion Coefficients
			P~C=(U0V01)T
		Matrix of Intrinsic Parameters	Known Optic Center
**Linear Initial Values Solving** **(RAC Constraints)**	$\left[\begin{array}{cccc} a_{1} & a_{2} & \cdots & a_{7} \end{array}\right]\to \left[\begin{array}{cccc} s_{x} & R_{3\times 3} & T_{x} & T_{y} \end{array}\right]$	$\left[\begin{array}{cc} T_{z} & f \end{array}\right]$	
	1st Step	2nd Step	
**Nonlinear Optimization**	$I(T_{z},\;f,\;k_{1},\;k_{2},\;k_{3})=\min$		
**Nonlinear Optimization** **(Binocular Camera System)**	$I(T_{a\_z},{\;f}_{a},\;k_{a1},\;k_{a2},\;k_{a3},T_{b\_z},{\;f}_{b},\;k_{b1},\;k_{b2},\;k_{b3})=\min$		

**Table 2 sensors-23-08466-t002:** The mathematical model and algorithm flow of Zhang’s coplanar calibration methods.

**Mathematical Model** **(Zhang’s Method)**	$\rho {\tilde{P}}_{I}=K_{\mathrm{Zhang}}\left[\begin{array}{cc} R_{3\times 3} & T_{3\times 1} \end{array}\right]{\tilde{P}}_{W}=H_{3\times 3}{\tilde{P}}_{W}$	$K_{\mathrm{Zhang}}=\left[\begin{array}{ccc} f_{x} & \gamma & U_{0}\\ 0 & f_{y} & V_{0}\\ 0 & 0 & 1 \end{array}\right]$	$D=(\begin{array}{ccccc} k_{1} & k_{2} & p_{1} & p_{2} & k_{3} \end{array})$
		Matrix of Intrinsic Parameters	Distortion Coefficients
**Linear Initial Values Solving**	$\left(H_{1\_3\times 3},\;H_{2\_3\times 3}\cdots H_{N\_3\times 3})\to B_{3\times 3}\to [f_{x}\;\;f_{y}\;\;\gamma \;\;U_{0}\;\;V_{0}]\to [R_{1\_\mathrm{vector}3\times 1}\;\;T_{1\_\mathrm{vector}3\times 1}\cdots R_{N\_\mathrm{vector}3\times 1}\;\;R_{N\_\mathrm{vector}3\times 1}\right]$		
**Nonlinear Optimization**	$I(K_{\mathrm{Zhang}},\;D,{\;R}_{1\_\mathrm{vector}3\times 1},{\;T}_{1\_\mathrm{vector}3\times 1},\;\cdots ,{\;R}_{N}{}_{\_\mathrm{vector}3\times 1},{\;T}_{N\_\mathrm{vector}3\times 1})=\min$		
**Nonlinear Optimization** **(Binocular Camera System)**	$I(K_{a\_\mathrm{Zhang}},{\;D}_{a},K_{b\_\mathrm{Zhang}},{\;D}_{b},{\;R}_{a-b}{}_{\_\mathrm{vector}3\times 1},{\;T}_{a-b\_\mathrm{vector}3\times 1},{\;R}_{a1\_\mathrm{vector}3\times 1},{\;T}_{a1\_\mathrm{vector}3\times 1},\cdots ,{\;R}_{\mathrm{aN}\_\mathrm{vector}3\times 1},{\;T}_{\mathrm{aN}\_\mathrm{vector}3\times 1})=\min$		

**Table 3 sensors-23-08466-t003:** The mathematical model and algorithm flow of ACC non-coplanar calibration method for monocular camera system.

**Mathematical Model** **(ACC Method)**	$\rho ({\tilde{P}}_{I}-{\tilde{P}}_{C})=K_{\mathrm{ACC}}\left[\begin{array}{cc} R_{3\times 3} & T_{3\times 1} \end{array}\right]\left[\begin{array}{cc} \eta & \beta \end{array}\right]{\tilde{P}}_{p}=M_{3\times 4}{\tilde{P}}_{p}$	$K_{\mathrm{ACC}}=\left[\begin{array}{ccc} f/s_{x} & 0 & 0\\ 0 & f & 0\\ 0 & 0 & 1 \end{array}\right]$	$D=\left(\begin{array}{ccccc} k_{1} & k_{2} & p_{1} & p_{2} & k_{3} \end{array}\right)$
			Distortion Coefficients
			P~C=(U0V01)T
		Matrix of Intrinsic Parameters	Known Optic Center
**Linear Initial Value Solving**	$M_{3\times 4}=\left[\begin{array}{cccc} m_{1} & m_{2} & m_{3} & m_{4}\\ m_{5} & m_{6} & m_{7} & m_{8}\\ m_{9} & m_{10} & m_{11} & 1 \end{array}\right]$		
**Nonlinear Optimization**	$I(m_{1},\;\cdots ,\;m_{11},\;k_{1},\;k_{2},\;p_{1},\;p_{2},\;k_{3})=\min$		
**Final Parameters Separation**	$\left(m_{1},\;\cdots ,\;m_{11}\right)\to K_{\mathrm{ACC}}=\left[\begin{array}{ccc} f/s_{x} & 0 & 0\\ 0 & f & 0\\ 0 & 0 & 1 \end{array}\right]\;\mathrm{and}\;\left[\begin{array}{cc} \eta & \beta \end{array}\right]=\left[\begin{array}{cccc} 1 & \eta_{x} & \beta_{x} & 0\\ 0 & \eta_{y} & \beta_{y} & 0\\ 0 & 0 & \beta_{z} & 0\\ 0 & 0 & 0 & 1 \end{array}\right]\;\mathrm{and}\;\left[\begin{array}{cc} R_{3\times 3} & T_{3\times 1} \end{array}\right]$		

**Table 4 sensors-23-08466-t004:** The mathematical model and algorithm flow of ACC non-coplanar calibration method for binocular camera system.

**Mathematical Model** **(ACC Method)** **(Extrinsic Parameters Calibration Only)**	$\left\{ \begin{array}{l} \rho_{a}({\tilde{P}}_{a\_I}-{\tilde{P}}_{a\_C})=K_{a\_\mathrm{ACC}}\left[\begin{array}{cc} R_{a\_3\times 3} & T_{a\_3\times 1} \end{array}\right]\left[\begin{array}{cc} \eta & \beta \end{array}\right]{\tilde{P}}_{p}=K_{a\_\mathrm{ACC}}G_{3\times 4}{\tilde{P}}_{p}\\ \rho_{b}({\tilde{P}}_{b\_I}-{\tilde{P}}_{b\_C})=K_{b\_\mathrm{ACC}}\left[\begin{array}{cc} R_{b\_3\times 3} & T_{b\_3\times 1} \end{array}\right]\left[\begin{array}{cc} \eta & \beta \end{array}\right]{\tilde{P}}_{p}=K_{b\_\mathrm{ACC}}H_{3\times 4}{\tilde{P}}_{p} \end{array}\right.$	$\left(K_{a\_\mathrm{ACC}},{\;K}_{b\_\mathrm{ACC}}\right)$	$\left(D_{a},{\;D}_{b}\right)$
			Calibrated Distortion Coefficients
			$\left({\tilde{P}}_{a\_C},{\tilde{P}}_{b\_C}\right)$
		Calibrated Intrinsic Parameters	Known Optic Centers
**Linear Initial Value Solving**	$G_{3\times 4}=\left[\begin{array}{cccc} g_{1} & g_{2} & g_{3} & g_{4}\\ g_{5} & g_{6} & g_{7} & g_{8}\\ g_{9} & g_{10} & g_{11} & 1 \end{array}\right]{\;\mathrm{and}\;H}_{3\times 4}=\left[\begin{array}{cccc} h_{1} & h_{2} & h_{3} & h_{4}\\ h_{5} & h_{6} & h_{7} & h_{8}\\ h_{9} & h_{10} & h_{11} & 1 \end{array}\right]$		
**Penalty Constraints Construction**	$\left(f_{p1},{\;f}_{p2}{,\;f}_{p3},{\;f}_{p4},{\;f}_{p5},{\;f}_{p6},{\;f}_{p7}\right)$		
**Nonlinear Optimization**	$I(g_{1},\;\cdots ,\;g_{11},\;h_{1},\;\cdots ,\;h_{11},{\;f}_{p1},\;\cdots ,{\;f}_{p7})=\min$		
**Final Parameters Separation**	$\left\{ \begin{array}{l} \left(g_{1},\;\cdots ,\;g_{11}\right)\\ \left(h_{1},\;\cdots ,\;h_{11}\right) \end{array}\right.\to \left[\begin{array}{cc} \eta & \beta \end{array}\right]=\left[\begin{array}{cccc} 1 & \eta_{x} & \beta_{x} & 0\\ 0 & \eta_{y} & \beta_{y} & 0\\ 0 & 0 & \beta_{z} & 0\\ 0 & 0 & 0 & 1 \end{array}\right]\;\mathrm{and}\;\left[\begin{array}{cc} R_{a\_3\times 3} & T_{a\_3\times 1} \end{array}\right]\;\mathrm{and}\;\left[\begin{array}{cc} R_{b\_3\times 3} & T_{b\_3\times 1} \end{array}\right]$		

**Table 5 sensors-23-08466-t005:** Proposed novel improved affine coordinate correction (IACC) mathematical model for non-coplanar calibration.

**Mathematical Model** **(Present Improved-ACC Method)**	$\rho ({\tilde{P}}_{I}-{\tilde{P}}_{C})=K_{\mathrm{IACC}}\left[\begin{array}{cc} R_{3\times 3} & T_{3\times 1} \end{array}\right]\left[\beta \right]{\tilde{P}}_{p}=A_{3\times 4}{\tilde{P}}_{p}$	$K_{\mathrm{IACC}}=\left[\begin{array}{ccc} f_{x} & \gamma & 0\\ 0 & f_{y} & 0\\ 0 & 0 & 1 \end{array}\right]$	$D=\left(\begin{array}{ccccc} k_{1} & k_{2} & p_{1} & p_{2} & k_{3} \end{array}\right)$
			Distortion Coefficients
			${\tilde{P}}_{C}={\left(\begin{array}{ccc} U_{0} & V_{0} & 1 \end{array}\right)}^{T}$
		Matrix of Intrinsic Parameters	Optic Center (Initial as Image Center)
**Linear Initial Value Solving**	$A_{3\times 4}=\left[\begin{array}{cccc} a_{1} & a_{2} & a_{3} & a_{4}\\ a_{5} & a_{6} & a_{7} & a_{8}\\ a_{9} & a_{10} & a_{11} & 1 \end{array}\right]\to K_{\mathrm{IACC}}=\left[\begin{array}{ccc} f_{x} & \gamma & 0\\ 0 & f_{y} & 0\\ 0 & 0 & 1 \end{array}\right]\;\mathrm{and}\;\left[\beta \right]=\left[\begin{array}{cccc} 1 & 0 & \beta_{x} & 0\\ 0 & 1 & \beta_{y} & 0\\ 0 & 0 & \beta_{z} & 0\\ 0 & 0 & 0 & 1 \end{array}\right]\;\mathrm{and}\;\left[\begin{array}{cc} R_{vector3\times 1} & T_{3\times 1} \end{array}\right]$		
**Nonlinear Optimization**	$I(K_{\mathrm{IACC}},{\;P}_{C}{\;,R}_{\mathrm{vector}3\times 1},{\;T}_{\mathrm{vector}3\times 1},\;D,\;\beta_{x},\;\beta_{y})=\min$		
**Nonlinear Optimization** **(Binocular Camera System)**	$I(K_{a-\mathrm{IACC}},{\;P}_{a-C}{\;,R}_{a-\mathrm{vector}3\times 1},{\;T}_{a-\mathrm{vector}3\times 1},{\;D}_{a},{\;K}_{b-\mathrm{IACC}},{\;P}_{b-C}{\;,R}_{b-\mathrm{vector}3\times 1},{\;T}_{b-\mathrm{vector}3\times 1},{\;D}_{b},\;\beta_{x},\;\beta_{y})=\min$		

**Table 6 sensors-23-08466-t006:** The Information of the Calibration Targets’ Pattern.

Pattern Type	Number of Columns	Number of Rows	Clearance of Feature Points
**Chessboard**	11	8	10 mm ± 1 μm ^1^
**Symmetric Circles**	9	7	15 mm ± 1 μm ^1^

^1^ This level of accuracy is decided by the manufacturing technique.

**Table 7 sensors-23-08466-t007:** Parameters of Chessboard Data Sets made by findChessboardCorners.

Data Sets	Number of Feature Point-Pairs	Applied Methods	Type
**Single_L_Chessboard_Laser_V20**	1760	**Tsai’s Method, ACC and IACC Method**	Non-Coplanar
**Single_R_Chessboard_Laser_V20**	1760	**Tsai’s Method, ACC and IACC Method**	Non-Coplanar
**Single_L_Chessboard_ZZY**	1760	**Zhang’s Method**	Coplanar
**Single_R_Chessboard_ZZY**	1760	**Zhang’s Method**	Coplanar

**Table 8 sensors-23-08466-t008:** This table shows the Single_L camera calibration results using the chessboard pattern with feature points found by findChessboardCorners from OpenCV 3.3.0.

Pattern Type (Extraction Algorithm)	Chessboard (findChessboardCorners)			
Camera	Single_L Camera			
Methods	Tsai’s	ACC Method	Zhang’s	IACC Method
**(*f_x_*, *f_y_*)**	(2251.457, 2251.750)	(2252.358, 2252.167)	(2254.887, 2255.056)	(2254.071, 2254.118)
**(*U*_0_, *V*_0_)**	(640, 512)	(650, 556)	(647.79, 493.60)	(639.54, 487.16)
**(*k*_1_, *k*_2_)**	(1.23 × 10^−8^, −9.02 × 10^−15^)	(1.16 × 10^−8^, −7.38 × 10^−15^)	(−6.47 × 10^−2^, 2.70 × 10^−1^)	(−7.29 × 10^−2^, 4.34 × 10^−1^)
**(*p*_1_, *p*_2_)**	-	(−2.62 × 10^−7^, 2.63 × 10^−9^)	(1.17 × 10^−3^, 5.35 × 10^−4^)	(1.40 × 10^−3^, 7.26 × 10^−4^)
**Reprojection Error * (Unit: Pix)**	**0.641**	**0.084**	**0.177**	**0.074**
**Reprojection Error * (Unit: mm)**	**0.059**	**0.008**	**0.019**	**0.007**

* Reprojection error (bold data in the table) is the key data to evaluate the calibration accuracy.

**Table 9 sensors-23-08466-t009:** This table shows the Single_R camera calibration results using the chessboard pattern with feature points found by findChessboardCorners from OpenCV 3.3.0.

Pattern Type (Extraction Algorithm)	Chessboard (findChessboardCorners)			
Camera	Single_R Camera			
Methods	Tsai’s	ACC Method	Zhang’s	IACC Method
**(*f_x_*, *f_y_*)**	(2252.373, 2252.480)	(2246.971, 2246.647)	(2247.280, 2247.239)	(2246.966, 2246.797)
**(*U*_0_, *V*_0_)**	(640, 512)	(689, 514)	(674.23, 505.23)	(675.70, 511.10)
**(*k*_1_, *k*_2_)**	(1.34 × 10^−8^, −3.51 × 10^−15^)	(1.55 × 10^−8^, −8.62 × 10^−15^)	(−7.39 × 10^−2^, 1.95 × 10^−1^)	(−6.94 × 10^−2^, 2.85 × 10^−2^)
**(*p*_1_, *p*_2_)**	-	(−5.95 × 10^−7^, 3.66 × 10^−8^)	(1.52 × 10^−4^, 1.48 × 10^−3^)	(−1.52 × 10^−5^, 1.45 × 10^−3^)
**Reprojection Error * (Unit: Pix)**	**0.539**	**0.070**	**0.098**	**0.066**
**Reprojection Error * (Unit: mm)**	**0.049**	**0.006**	**0.010**	**0.006**

* Reprojection (bold data in the table) error is the key data to evaluate the calibration accuracy.

**Table 10 sensors-23-08466-t010:** This table shows the Single_L camera calibration results using the symmetric circle pattern with feature points found by findCirclesGrid from OpenCV 3.3.0.

Pattern Type (Extraction Algorithm)	Symmetric Circle Pattern (findCirclesGrid)			
Camera	Single_L Camera			
Methods	Tsai’s	Zhang’s	ACC Method	IACC method
**(*f_x_*, *f_y_*)**	(2258.761, 2258.832)	(2258.761, 2258.832)	(2253.180, 2253.028)	(2254.551, 2254.486)
**(*U*_0_, *V*_0_)**	(640, 512)	(646.36, 493.89)	(646, 541)	(639.47, 478.33)
**(*k*_1_, *k*_2_)**	(1.17 × 10^−8^, −8.91 × 10^−15^)	(−5.75 × 10^−2^, 2.39 × 10^−1^)	(1.10 × 10^−8^, −7.60 × 10^−15^)	(−6.61 × 10^−2^, 3.38 × 10^−1^)
**(*p*_1_, *p*_2_)**	-	(1.32 × 10^−3^, 4.50 × 10^−4^)	(−2.53 × 10^−7^, −7.23 × 10^−9^)	(1.42 × 10^−3^, 6.62 × 10^−4^)
**Reprojection Error * (Unit: Pix)**	**0.629**	**0.089**	**0.070**	**0.024**
**Reprojection Error * (Unit: mm)**	**0.064**	**0.011**	**0.007**	**0.002**

* Reprojection error (bold data in the table) is the key data to evaluate the calibration accuracy.

**Table 11 sensors-23-08466-t011:** This table shows the Single_R camera calibration results using the symmetric circle pattern with feature points found by findCirclesGrid from OpenCV 3.3.0.

Pattern Type (Extraction Algorithm)	Symmetric Circle Pattern (findCirclesGrid)			
Camera	Single_R camera			
Methods	Tsai’s	Zhang’s	ACC Method	IACC method
**(*f_x_*, *f_y_*)**	(2232.061, 2231.872)	(2245.017, 2245.000)	(2244.275, 2244.229)	(2244.373, 2244.342)
**(*U*_0_, *V*_0_)**	(640, 512)	(670.80, 503.99)	(659, 494)	(664.59, 491.88)
**(*k*_1_, *k*_2_)**	(8.60 × 10^−9^, 3.95 × 10^−15^)	(−5.62 × 10^−2^, 2.03 × 10^−1^)	(1.53 × 10^−8^, −1.06 × 10^−14^)	(−8.26 × 10^−2^, 3.93 × 10^−1^)
**(*p*_1_, *p*_2_)**	-	(1.41 × 10^−4^, 1.42 × 10^−3^)	(−7.54 × 10^−7^, −5.65 × 10^−9^)	(5.22 × 10^−5^, 1.58 × 10^−3^)
**Reprojection Error * (Unit: Pix)**	**1.034**	**0.044**	**0.042**	**0.041**
**Reprojection Error * (Unit: mm)**	**0.109**	**0.005**	**0.004**	**0.004**

* Reprojection error (bold data in the table) is the key data to evaluate the calibration accuracy.

**Table 12 sensors-23-08466-t012:** This table shows the Single_L camera calibration results using symmetric circle patterns with feature points found by Present New Algorithm in Appendix B.

Methods	Zhang’s		Present IACC Method	
Feature Extraction Algorithm	findCirclesGrid	New Method in Appendix B	findCirclesGrid	New Method in Appendix B
**(*f_x_*, *f_y_*)**	(2258.761, 2258.832)	(2257.609, 2257.477)	(2254.551, 2254.486)	(2254.686, 2254.609)
**(*U*_0_,** ** *V* _0_ ** **)**	(646.36, 493.89)	(647.60, 493.95)	(639.472, 478.326)	(639.152, 480.773)
**(*k*_1_, *k*_2_)**	(−5.75 × 10^−2^, 2.39 × 10^−1^)	(−5.67 × 10^−2^, 2.37 × 10^−1^)	(−6.61 × 10^−2^, 3.38 × 10^−1^)	(−6.14 × 10^−2^, 2.72 × 10^−1^)
**(*p*_1_, *p*_2_)**	(1.32 × 10^−3^, 4.50 × 10^−4^)	(1.26 × 10^−3^, 6.62 × 10^−4^)	(1.42 × 10^−3^, 6.62 × 10^−4^)	(1.45 × 10^−3^, 6.57 × 10^−4^)
**Reprojection Error * (Unit: Pix)**	**0.089**	**0.035**	**0.024**	**0.019**
**Reprojection Error * (Unit: mm)**	**0.011**	**0.004**	**0.002**	**0.002**

* Reprojection error (bold data in the table) is the key data to evaluate the calibration accuracy.

**Table 13 sensors-23-08466-t013:** This table shows the Single_R camera calibration results using symmetric circle pattern with feature points found by the Present New Algorithm in Appendix B.

Methods	Zhang’s		Present IACC Method	
Feature Extraction Algorithm	findCirclesGrid	New Method in Appendix B	findCirclesGrid	New Method in Appendix B
**(*f_x_*, *f_y_*)**	(2245.017, 2245.000)	(2258.622, 2258.425)	(2244.373, 2244.342)	(2243.868, 2243.764)
**(*U*_0_, *V*_0_)**	(670.80, 503.99)	(671.442, 504.214)	(664.590, 491.874)	(663.304, 501.017)
**(*k*_1_, *k*_2_)**	(−5.62 × 10^−2^, 2.03 × 10^−1^)	(−5.78 × 10^−2^, 2.34 × 10^−1^)	(−8.26 × 10^−2^, 3.93 × 10^−1^)	(−7.11 × 10^−2^, 1.99 × 10^−1^)
**(*p*_1_, *p*_2_)**	(1.41 × 10^−4^, 1.42 × 10^−3^)	(1.59 × 10^−4^, 1.54 × 10^−4^)	(5.22 × 10^−5^, 1.58 × 10^−3^)	(5.82 × 10^−5^, 1.59 × 10^−3^)
**Reprojection Error * (Unit: Pix)**	**0.044**	**0.033**	**0.041**	**0.028**
**Reprojection Error * (Unit: mm)**	**0.005**	**0.004**	**0.004**	**0.003**

* Reprojection error (bold data in the table) is the key data to evaluate the calibration accuracy.

**Table 14 sensors-23-08466-t014:** This table shows the calibration results separately calibrated by the ACC method, Zhang’s method, and the present IACC method.

Pattern Type		Symmetric Circle Pattern (Present New Algorithm in Appendix B)			
**Number of Images-Pairs**		**5**			
**Methods**		**ACC Method**		**Zhang’s Method**	
**Type**		Non-coplanar		Coplanar	
** *K_a_* **	** *K_b_* **	$\left[\begin{array}{ccc} 2253.180 & 0 & 646\\ 0 & 2253.027 & 541\\ 0 & 0 & 1 \end{array}\right]$	$\left[\begin{array}{ccc} 2244.275 & 0 & 659\\ 0 & 2244.229 & 494\\ 0 & 0 & 1 \end{array}\right]$	$\left[\begin{array}{ccc} 2249.114 & 0 & 646.522\\ 0 & 2248.951 & 492.704\\ 0 & 0 & 1 \end{array}\right]$	$\left[\begin{array}{ccc} 2242.060 & 0 & 675.820\\ 0 & 2241.410 & 504.346\\ 0 & 0 & 1 \end{array}\right]$
** *D_a_* **	** *D_b_* **	$\left[\begin{array}{ccccc} {1\times 10}^{-8} & {-7\times 10}^{-15} & 6\times 10^{-14} & {-3\times 10}^{-7} & {-7\times 10}^{-9} \end{array}\right]$	$\left[\begin{array}{ccccc} {1\times 10}^{-8} & {-1\times 10}^{-14} & 8{\times 10}^{-15} & {-8\times 10}^{-7} & {-6\times 10}^{-9} \end{array}\right]$	$\left[\begin{array}{ccccc} -0.067 & 0.374 & 0.001 & 0.001 & 0.803 \end{array}\right]$	$\left[\begin{array}{ccccc} -0.067 & 0.145 & {4\times 10}^{-5} & 0.001 & 0.640 \end{array}\right]$
** *β* **	** *η_x_* **	${\left[\begin{array}{cc} {2.772e}^{-3} & {1.678e}^{-4} \end{array}\right]}^{T}$	−3.133*e*^−3^	-	
** *R_a_b_Mat_* ** ** _3×3_ **		$\left[\begin{array}{ccc} 0.861 & {-6.030\times 10}^{-3} & 0.508\\ 0.005 & 1 & -0.004\\ -0.508 & -0.001 & 0.861 \end{array}\right]$		$\left[\begin{array}{ccc} 0.862 & -0.001 & 0.506\\ 0.002 & 1 & -0.001\\ -0.506 & 0.002 & 0.862 \end{array}\right]$	
** *T_a_b_* **		${\left[\begin{array}{ccc} -121.089 & 4.780 & 32.076 \end{array}\right]}^{T}$		${\left[\begin{array}{ccc} -121.776 & -0.226 & 31.857 \end{array}\right]}^{T}$	
**Reprojection Error ***		**0.594 Pixels**		**0.055 Pixels**	
**Methods**		**Present IACC Method**			
**Type**		Non-coplanar			
** *K_a_* **	** *K_b_* **	$\left[\begin{array}{ccc} 2257.200 & 1.626 & 643.293\\ 0 & 2255.697 & 487.601\\ 0 & 0 & 1 \end{array}\right]$		$\left[\begin{array}{ccc} 2241.096 & -1.633 & 667.915\\ 0 & 2242.470 & 498.817\\ 0 & 0 & 1 \end{array}\right]$	
** *D_a_* **	** *D_b_* **	$\left[\begin{array}{ccccc} -0.062 & 0.298 & 0.001 & {7\times 10}^{-4} & -0.392 \end{array}\right]$		$\left[\begin{array}{ccccc} -0.066 & 0.125 & {2\times 10}^{-4} & 0.001 & -0.616 \end{array}\right]$	
** *β* **		${\left[\begin{array}{cc} {2.772\times 10}^{-3} & {1.678\times 10}^{-4} \end{array}\right]}^{T}$			
** *R_a_b_Mat_* ** ** _3×3_ **		$\left[\begin{array}{ccc} 0.863 & {-2.215\times 10}^{-4} & 0.506\\ 0.001 & 1 & -0.001\\ -0.506 & 0.002 & 0.863 \end{array}\right]$			
** *T_a_b_* **		${\left[\begin{array}{ccc} -121.884 & -0.100 & 31.421 \end{array}\right]}^{T}$			
**Reprojection Error ***		**0.032 Pixels**			

* Reprojection error in Pixels (bold data in the table) is a key data to evaluate the calibration accuracy of different methods.

**Table 15 sensors-23-08466-t015:** This table shows the 3D reconstruction error data of the present calibration method and Zhang’s method with the target in mono-viewing multiplane position.

Clearances Type	Horizontal Point-Pairs Clearances Error Data			Vertical Point-Pairs Clearances Error Data		
**Number of Measured Clearances**	**280**			**270**		
**Real Clearance Value**	**15 mm ± 1 μm**			**15 mm ± 1 μm**		
**Calibration Methods**	**Zhang’s**	**ACC**	**IACC**	**Zhang’s**	**ACC**	**IACC**
**Reproject Error of Calibration**	**0.055 Pixels** **(Non-Measured Position)**	**0.594 Pixels** **(Measured Position)**	**0.032 Pixels** **(Measured Position)**	**0.055 Pixels** **(Non-Measured Position)**	**0.594 Pixels** **(Measured Position)**	**0.032 Pixels** **(Measured Position)**
**Measured Average Value (Unit: mm)**	14.9510	14.9952	15.0003	14.9488	15.0139	15.0001
**Abs of Average Error (Unit: mm)**	0.0490	0.0048	0.0003	0.0512	0.0139	0.0001
**Root Mean Square Error** * **(Unit: mm)**	**0.0520**	**0.0211**	**0.0026**	**0.0521**	**0.0300**	**0.0018**

* RMS error (bold data in the table) is the key data to evaluate the accuracy of distance measurements.

**Table 16 sensors-23-08466-t016:** This table shows the 3D reconstruction error data of the present calibration method and Zhang’s method with the target in multi-viewing multiplane position.

Clearances Type	Horizontal Point-pairs Clearances Error Data			Vertical Point-pairs Clearances Error Data		
**Number of Measured Clearances**	**280**			**270**		
**Real Clearance Value**	**15 mm ± 1 μm**			**15 mm ± 1 μm**		
**Calibration Methods**	**Zhang’s**	**ACC**	**IACC**	**Zhang’s**	**ACC**	**IACC**
**Reproject Error of Calibration**	**0.055 Pixels** **(Measured Position)**	**0.594 Pixels** **(Non-Measured Position)**	**0.032 Pixels** **(Non-Measured Position)**	**0.055 Pixels** **(Measured Position)**	**0.594 Pixels** **(Non-Measured Position)**	**0.032 Pixels** **(Non-Measured Position)**
**Measured Average Value (Unit: mm)**	15.0076	15.0508	15.0569	15.0077	15.0694	15.0521
**Abs of Average Error (Unit: mm)**	0.0076	0.0508	0.0569	0.0077	0.0694	0.0521
**Root Mean Square Error** * **(Unit: mm)**	**0.0217**	**0.0575**	**0.0591**	**0.0173**	**0.0770**	**0.0544**

* RMS error (bold data in the table) is the key data to evaluate the accuracy of distance measurements.

**Table 17 sensors-23-08466-t017:** The fitting results of local ROIs of three cylinders with different radii.

**Calibration** **Parameters from IACC method**	**Fitting Results**	$\left(x_{0},y_{0},z_{0}\right)$	(l,m,n)	*r*	**Numbers of Points**	**RMSE**
	**Cylinder #1**	(77.168, −1.588, 246.656)	(−0.003, 1.000, −0.001)	76.369	150,000	0.0065
	**Cylinder #2**	(65.792, 0.095, 290.116)	(0.009, 1.000, −0.007)	103.510	100,000	0.0099
	**Cylinder #3**	(52.377, −14.842, 302.689)	(0.077, 2.692, −0.076)	124.870	150,000	0.0063
**Calibration** **Parameters from Zhang’s Method**	**Fitting Results**	$\left(x_{0},y_{0},z_{0}\right)$	(l,m,n)	*r*	**Numbers of Points**	**RMSE**
	**Cylinder #1**	(75.656, 1.517, 246.340)	(−0.003, 1.000, −0.001)	76.704	150,000	0.0066
	**Cylinder #2**	(64.074, 0.198, 289.784)	(0.010, 1.000, −0.007)	103.980	100,000	0.0100
	**Cylinder #3**	(50.785, −8.227, 302.300)	(0.029, 1.000, −0.028)	125.469	150,000	0.0062

**Table 18 sensors-23-08466-t018:** Quantified complexity of the mentioned four calibration methods.

Methods	Tsai		Zhang		ACC		IACC	
Calibration Mode	Monocular	Binocular	Monocular	Binocular	Monocular	Binocular	Monocular	Binocular
**Complexity of** **Algorithm**	5.04 × 10^5^	1.008 × 10^6^	1.7388 × 10^7^	3.477 × 10^7^	3.780 × 10^6^	1.5120 × 10^7^	4.284 × 10^6^	8.568 × 10^6^
**Complexity of** **Implementation**	3 × 10^8^	3 × 10^8^	1 × 10^8^	1 × 10^8^	1.5 × 10^8^	3 × 10^8^	1.5 × 10^8^	1.5 × 10^8^

## Data Availability

All data that support the findings of this study are included within this article.

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
