# Peer review of "A Stable, Efficient, and High-Precision Non-Coplanar Calibration Method: Applied for Multi-Camera-Based Stereo Vision Measurements"

_sensors, 2023, doi:10.3390/s23208466_

Round 1
Reviewer 1 Report
The authors have introduced a new and accurate scheme for camera calibration, which may be informative to the readers of this journal. But the structure of the manuscript must be improved drastically.
First of all, the manuscript is too lengthy and is not well organized.
The reviewer recommends the authors to rewrite the whole manuscript by focusing on the mathematical model of camera calibration only and moving the circle detection method to Appendix while removing the current Appendices.
As for the description of this manuscript, the naming of four methods should be improved so that the reader can understand the characteristics of each method. Tsai's method and Zhang's methods are OK with their name, or Tsai's method can be called "off-the-shelf camera calibration scheme" or "pinhole camera model". Zhang's method might be “pinhole camera model including skew of two image axes”.
The confusions by reader may arise by "Zheng-GN method" and "Paper's Method". What does "GN" stand for? Gauss-Newton? It's not appropriate to name the paper [8] by Gauss-Newton, but the reviewer recommends the author to name it as such as "ACC(Affine Coordinate Correction) method" or something else. The reviewer also recommends the authors to name their own method by its mathematical characteristics. Furthermore, as some of the authors of Zheng-GN method and Paper's method are the same, the reviewer thinks that both methods are "(authors') our methods". Then Zheng-GN method can be called as Zheng's method or by "our method" and Paper's method can be called for example "Zheng's method with SOME improvement".
Another confusion with this manuscript may arise from the treatments about monocular and binocular calibration. As the authors have pointed out in line 1207-1212, Zheng-GN method involves the relationship between 2 cameras, but the reviewer thinks that the discussion should be limited by single camera calibration.
The reviewer thinks that the most important aspect of this study is in the mathematical treatment of the camera model. Thus, Figs. 2-4, 7 is not necessary but the mathematical description of all the 4 model should be addressed such as Eqs. (1)-(16).
As for the writing, the abbreviations such as "DLT", "RAC", "ROI", "DIC", "ICGN", "w.r.t." are not appropriate. The reviewer thinks that only the well-known technical terms can be used such as FOV, DOF, etc.
------------------------------------------------------------
The followings are the comment for individual descriptions.
The introduction of gamma in Eq. (6) and in line 460 is very important, but thinking about the current level of the manufacturing accuracy of the image sensors it can be easily assumed to be zero. It may be true that the small value of gamma in Table 9 did a great job in the present measurement, but it does not come from the skewness of the targets' two axes. The reviewer think that this value has worked well in compensating the orthogonality of the traverse direction to the targets' plane. Please add a physical explanation about the skewness of image axes.
Is it OK for the distortions in Eq. (9) to be added? The reviewer thinks that they should be multiplied.
BTW, did you check the flatness of the calibration plate? As the orthogonality of the traverse direction to the calibration plate may affect the calibration accuracy, a small curvature of the calibration plate also deteriorates the calibration results. As for the orthogonality of the traverse direction to the calibration plate, the reviewer always uses a laser pointer, which is carefully placed parallel to the traverse mechanism, to be reflected by a mirror pasted onto the calibration plate so that the reflected beam comes back to the laser pointer itself.
As for the patterns of calibration plate, even though the reviewer also recommends the symmetric circles grid pattern, the accuracy by chessboard pattern can be improved by slightly rotating the chessboard. When the chessboard grids are parallel to the pixels, the discretizations along the edge such as in Fig. 11 may not take the random values and the averaging will not improve the accuracy.
As for Fig. 15, the left top corner and the left bottom corner of the image sensor does not contain the calibration circles. It means that those regions of the image sensor are not calibrated. As the authors wrote in line 1135, all the points in the FOV should be covered by calibration plate. It implies that a larger matrix of circles should be used for calibration.
Simulations about Figs. 16-18 are helpful for the readers.
The comparisons between 4 methods as in Tables 3-7 are comprehensive but the point extraction method should be limited to the best case of Paper's Novel Algorithm, which should also be named such as "radial section scanning method".
Figs. 18-19 are not necessary.
Table 9 is most impressive of this manuscript. The authors should pay more attention to emphasize this result as the conclusion of this manuscript.
Even though the 3D measurements are done in real space and it can be understood that the results in Table 10-11 are expressed in mm, the accuracy of the calibration method should be characterized by pixels. The reviewer recommends the author to add the typical value in pixels to Table 10-11. Moreover, it is a disaster for the image calibration to have a non-zero value for average error. The authors should present the average error, while average absolute error and maximum absolute error are not necessary. BTW, Paper’s method had a great advantage for mono-viewing in Table 10, but the improvement is very small for multi-viewing in Table 11. It discourages the reader of this manuscript to use this method. Don’t you have any comments on this?
Is there any other conclusion about Figs. 27? The authors may be able to calculate the radii of curvature of each cylinder.
Ref. [xxx] is missing in line 812.
Overall, English is fine, but several sentences should be corrected. The reviewer recommends the authors to consult an English native speaker to check the whole manuscript.
Reviewer 2 Report
Authors propose a novel stable, efficient, and accurate non-coplanar calibration method which can be well applied for camera-based stereo vision measurements. This method can significantly correct existing non-coplanar calibration methods’ weakness, i.e., cumbersome operation, insufficient accuracy, unstable fitting different measurement scenes, etc. Experiments verify that the novel calibration methods are easy to operate, and have better accuracy than several classical methods. The manuscript provides a detailed description of the proposed method and provides detailed experimental results. The manuscript has reference value for practical applications. Therefore, I recommend publishing it on sensors. I also suggest reducing the length of the manuscript appropriately.
Reviewer 3 Report
In this manuscript, the authors propose a stable, efficient, and high-precision non-coplanar calibration method based on a novel calibration mathematical model and a simple circular feature point extraction algorithm. The feasibility of the proposal is evaluated by several simulations and experiments. It is suggested that the authors explain or revise the following questions:
1. From the content of the manuscript, it can be observed that its two major contributions are as follows: firstly, the establishment of a novel mathematical model to describe the transformation process from the 3D affine coordinates of targets to camera image coordinates. And a new non-coplanar calibration method suitable for both monocular and binocular camera systems is proposed based on this foundation. Secondly, the introduction of an innovative circular feature point extraction method, which is based on the region-based Otsu algorithm and the radial section scanning method, aims to achieve precise extraction of feature points. However, at the end of the introduction, the authors summarize four contributions, which may seem verbose. Please consider making revisions.
2. The second part of the content occupies a substantial portion of the manuscript. However, many of the descriptions in it are unnecessary, and there are some unclear descriptions. For example, when comparing the traditional Zhang’s method with Tsai’s method, it is mentioned that Zhang’s method requires solving too many parameters. Specific details about which types or which parameters should be solved for should be provided here. Additionally, the title of Figure 7 is incorrect; please correct it.
3. Please supplement the description of the symbol ρ in Eq. (6).
4. Please explain what 'Rodrigues' means in Eq. (15).
5. The mention of Figures 24(d), (e), and (f) can be found on line 1112. However, their corresponding images cannot be located in the manuscript. Please make the necessary corrections.
6. This manuscript needs to be rewritten both in terms of language expression and content structure.
In conclusion, the proposed non-coplanar calibration method is somewhat innovative by combining the novel mathematical model with circular feature point extraction algorithm to overcome the instability and inaccuracy problem of calibration methods in stereo vision measurements. However, I cannot recommend this manuscript for publication until the above issues are either addressed or adequately explained.

This manuscript needs to be rewritten both in terms of language expression and content structure.
Reviewer 4 Report
This article proposes a mathematical model to represent the transformation from three-dimensional affine coordinates to camera image coordinates based on projective geometry and matrix transformation theory. This model proposes a non-coplanar camera calibration method for monocular or binocular visual systems. Additionally, a circular feature point extraction method is proposed to enhance the accuracy and stability of the calibration method. According to the elaboration in the paper and related experiments, some progress has been made in this field. However, the author should consider several aspects:
1. The paper should be shorter. It mainly includes the following four points:
a. Section II needs to be longer and provide a comprehensive review of relevant work, focusing only on Zhang and Tsai's methods and ignoring contributions from other scholars in this field. The author should provide a more thorough and concise literature review rather than selecting one or two long articles to elaborate on.
b. Some parts of Section II, such as Section 5, could be placed later.
c. The paper contains a large number of formulas. The reasoning process of formulas (2) to formulas (18) takes up much space. This reasoning process merely adds a coordinate transformation to the camera imaging model, and the rest of the reasoning steps and calculation results are similar to those of the original camera imaging model. There is no need to elaborate on the imaging process in a lengthy manner. It is recommended to provide additional materials or simplify the reasoning process.
d. The Conclusions section of the paper should be appropriately streamlined, focusing on presenting one's results. Many similar points are not listed here, but the paper could be more concise and organized, and it should consider deleting or condensing most of its content.
2. The paper needs some essential content. It mainly includes the following four points:
a. The paper needs a review of research in recent years in this field. Appropriate supplementary material should be added.
b. The paper needs more necessary discussions instead of elaborating on experimental results. Why is your method effective? What factors affect stability? What measures have you taken to address these factors and achieve good results? These effects have been proven by which experiments? Because of the organization of the paper and the large amount of information crammed, it is not easy to find answers to several questions I have mentioned.
c. How was the reduction in computational complexity proved? Can you provide a more precise estimate of the computational complexity of your proposed algorithm?
d. The paper needs more aesthetic appeal. This may not be a severe issue, but a neat, concise layout with consistent formatting and images would be more conducive to us reviewers getting the information we want.
In summary, although the paper is built upon cramming, it lacks specific information, or the lengthy layout and length have covered up the true essence of the paper.
3. I need some clarification about the method proposed in the paper. It mainly includes two points:
a. The camera imaging model is easy to understand, but formula (1) adds a coordinate transformation on top of the camera imaging model, which is considered innovative and hard to comprehend. From left to right, formula (1) is merely a primary transformation (the basis for linear transformations) combined with rotation and translation matrices into a new rotation and translation matrix. Although two additional items are in the formula (15) compared to the original camera model's uncalibrated result, these two items can be regarded as a contribution extracted from the new rotation and translation matrix. Your subsequent experiments have yet to prove the impact of these two additional items on experimental results, and a clear explanation is expected.
b. Sobel algorithm, ROI, etc., are new algorithms? The combination of ROI with the gradient method is also not original. You may add some necessary references. Moreover, it describes the algorithm in sufficient detail and focuses on self-innovation.
Grammar errors, poor formatting, etc., affect the aesthetic appeal of the paper. More consideration should be given to these issues as they affect the paper's readability.
Round 2
Reviewer 1 Report
Even though the authors have agreed to all my recommendations, the authors have amended their manuscript in a patchwork manner. The reviewer recommends the authors to read through their new manuscript carefully, so that ALL the sentences contribute to the single story of the manuscript.
The followings are comments to the individual descriptions:
The authors introduce the 4 camera models in the order of Zhang’s in 2.1 and then Tsai’s, ACC and IACC in 2.2. The reviewer recommends the authors to align them in historical order.
In 2.4, the authors still discuss about coplanar calibration. But the reviewer recommends the authors to remove Fig. 3 to make the manuscript concise.
As the authors moved the discussion about ROI to Appendix A, 2.5 should also be moved to Appendix A.
2.6 should be merged with 5.6.
Even though the authors say in line 489 that the explanation about gamma will be done in Section 4.1, there is no description about gamma in Section 4.1.
Is the title of Section 4 correct? ROI is not necessary. This is the evidence that the amendment by the authors is just a patchwork.
In Table 10 and 11, the order of 4 methods should be Tsai’s, Zhang’s, ACC and IACC. Is there any reason why the improvement by IACC is small for Single_R camera? The results for Single_L camera are impressive while those of Single_R camera are subtle. The authors SHOULD clearly state the cause of this difference, or the experiment for Single_R to be replaced by new data.
In Tables 12 and 13, “Paper’s IACC Method” should be replaced by “(present) IACC method”. “Paper’s Novel Algorithm” should be replaced by “our new method in Appendix A”. Several other descriptions of “Paper’s” are seen and should be replaced by “present”.
Is it true that Zhang's Method is Coplanar in Table 14?
BTW, what is “virtual strain gauge of xx-direction” in Line 1034?
In Table 15, IACC has a certain advantage over Zhang's and ACC foro mono-view, but there is not for multi-view in Table 16. Please explain the reason why.
Are there any difference in Figs. (a)-(c) and Figs. (d)-(f)?
In Table 17, the authors call the Cylinder by 1#, 2# and 3#, but is should be #1, #2 and #3. The results in Table 17 should also be compared with different camera models.
What is 1e^8 in line 1123? It’s just an Excel expression and is not mathematically correct.
Overall, English is fine, but several sentences should be corrected. The reviewer recommends the authors to consult an English native speaker to check the whole manuscript.
Reviewer 3 Report
The author has answered the questions raised by the reviewers one by one, and the inappropriate language expression and content structure have been revised in the manuscript. The revised manuscript basically meets the requirements of the reviewers, and can be accepted.
Reviewer 4 Report
My confusion has been answered, but I still hope that the author will control the length of the article, consider parts of the content as supplementary material, and not appear in the main text.
Nothing.
